

# A finite-element framework to explore the numerical solution of the coupled problem of heat conduction, water vapor diffusion and settlement in dry snow (IvoriFEM v0.1.0)

Julien Brondex[1], Kévin Fourteau[1], Marie Dumont[1], Pascal Hagenmuller[1], Neige Calonne[1], François Tuzet[1], and Henning Löwe[2]

[1]Univ. Grenoble Alpes, Université de Toulouse, Météo-France, CNRS, CNRM, Centre d'Études de la Neige, 38000 Grenoble, France
[2]WSL Institute for Snow and Avalanche Research SLF, Flüelastr. 11, 7260 Davos, Switzerland

**Correspondence:** Julien Brondex (julien.brondex@meteo.fr)

**Abstract.** The poor treatment, or complete omission, of water vapor transport has been identified as a major limitation suffered by currently available snowpack models. Vapor and heat fluxes being closely intertwined, their mathematical representation amounts to a system of non-linear and tightly-coupled partial differential equations, which is particularly challenging to solve numerically. The choice of the numerical scheme and the representation of couplings between processes is crucial to ensure

an accurate and robust solution that guarantees mass and energy conservation, while allowing time steps in the order of 15 minutes. To explore the numerical treatments fulfilling these requirements, we have developed a highly-modular finite-element program. The code is written in python. Every step of the numerical formulation and solution is coded internally, except for the inversion of the linearized system of equations. We illustrate the capabilities of our approach to tackle the coupled problem of heat conduction, vapor diffusion and settlement within a dry snowpack by running our model on several test cases proposed

in recently published literature. We underline specific improvements regarding energy and mass conservation, as well as time step requirements. In particular, we show that a fully-coupled and fully-implicit time stepping approach enables to get accurate and stable solutions with little restriction on the time step.

## 1 Introduction

Over the last decades, snow models of various complexity have been developed for a myriad of applications, including avalanche forecasting (Morin et al., 2020), water resources management (Magnusson et al., 2015), glacier mass balance assessment (e.g., van Pelt et al., 2012; Sauter et al., 2020), or projections of future climate evolution (Krinner et al., 2018). They range from single-layer snow schemes to detailed snowpack models providing an explicit description of the vertical distribution of physical properties, such as the models Crocus (Brun et al., 1989, 1992) and SNOWPACK (Bartelt and Lehning, 2002).

However, even the most detailed snow models suffer from major weaknesses (Menard et al., 2021). Their inability to reproduce





inverted density gradients as observed in Arctic snowpacks (Domine et al., 2016; Barrere et al., 2017), where strong tempera-ture gradients induce significant water vapor flux redistributing ice mass from basal to upper layers, has been pinpointed as one of those (Domine et al., 2019). More generally, vapor transport is involved in many processes, such as redistribution of water vapor isotopes in polar firn (Touzeau et al., 2018) or snow metamorphism (e.g., Sturm and Benson, 1997), with implications

on snowpack stability (e.g., Pfeffer and Mrugala, 2002). To address this need, efforts have been made in two directions: (i) the implementation of ad hoc water vapor transfer modules in existing models (e.g., Touzeau et al., 2018; Jafari et al., 2020), and (ii) the development of stand-alone models to explore the numerical treatment of the coupled heat and water vapor transfer problem (e.g., Simson et al., 2021; Schürholt et al., 2022).

Heat conduction and water vapor diffusion are two-way coupled: temperature gradients within the snowpack drive water
vapor fluxes; vapor fluxes transport latent heat which redistributes energy within the snowpack and feeds back on temperature gradients whenever sublimation/deposition occurs (Yosida et al., 1955; Sturm and Johnson, 1992; Albert and McGilvary, 1992; Pinzer et al., 2012). Furthermore, the microstructure evolves as water vapor deposits on or sublimates from the ice phase. This affects the effective thermal conductivity and effective vapor diffusion coefficient, which in turns echoes on the energy and vapor fluxes (Yosida et al., 1955; Jaafar and Picot, 1970; Sturm and Johnson, 1992; Calonne et al., 2011; Riche
and Schneebeli, 2013). Two mathematical descriptions of the macroscopic heat and water vapor budget in dry snow have been proposed. The first one was derived by Calonne et al. (2014) using the two-scale expansion method. The second one was introduced by Hansen and Foslien (2015) using mixture theory. Both models account for the interactions between energy and vapor fluxes through phase change. However, both models were derived on an invariant microstructure, thus neglecting the feedback of phase change on the distribution of the ice volume fraction, i.e. the fraction of the volume occupied by ice for a
given volume of snow.

While the heat equation is at the core of any snowpack model, and was therefore implemented at the very early stage of their decades-long developments (e.g., Brun et al., 1989; Bartelt and Lehning, 2002), efforts to include water vapor transfer are recent. Because it avoids in-depth modification of the code which is cumbersome and prone to bug dissemination (Menard et al., 2021), first order operator splitting is the most natural way to couple newly developed sub-processes to processes
already incorporated in previous versions of a model. Basically, this method consists in solving all sub-processes of a coupled process sequentially. Variants of this method have been used by Touzeau et al. (2018) and Jafari et al. (2020) to implement water vapor diffusion in, respectively, Crocus and SNOWPACK. However, this approach may require shorter time step than the main time step of the model. In the present case, the deposition/sublimation rate is controlled by the magnitude of the over/under-saturation of vapor which is highly temperature sensitive, and is modulated by the kinetics of absorption/desorption
of water molecules on the ice surfaces (Fourteau et al., 2021). It follows that solving the vapor mass balance and heat equations sequentially is prone to instabilities whenever the time step is too large relatively to the considered kinetics. Both Touzeau et al. (2018) and Jafari et al. (2020) have thus reported constraining time step requirements of, respectively, 1 s and 1 min. This is considerably shorter than the time step of 15 min used in Crocus and SNOWPACK for operational avalanche forecasting for example. Furthermore, some important feedback were not accounted for: (i) Touzeau et al. (2018) did not include any phase





change-related latent heat effects in the heat equation, and (ii) neither of the two studies considered the evolution of effective parameters (including viscosity) due to deposition/sublimation.

An alternative approach is to treat the heat conduction and water vapor diffusion as a single monolithic process. In two companion papers, Schürholt et al. (2022) and Simson et al. (2021) have developed their own stand-alone models to explore this path. More specifically, Schürholt et al. (2022) used the python-based finite element computing platform FEniCS in order to

solve both the models of Hansen and Foslien (2015) and of Calonne et al. (2014), while accounting for deposition/sublimation feedback on the ice volume fraction field. However, because remeshing was not supported in FEniCS at the time of the study, they did not include snow settlement. In contrast, Simson et al. (2021) have proposed a numerical approach which combines a deforming mesh procedure based on a Lagrangian method to solve for settlement, and a classical finite difference method applied on the deformed mesh to solve the model of Hansen and Foslien (2015) only. Both of these works constitute extremely

valuable contributions for the understanding of non-linear feedback. Yet, a number of limitations are left to be addressed: requirements regarding the time step are still not clearly established, and a careful assessment of mass and energy conservation is lacking.

Snowpack models are always said to be based on mass and energy conservation (e.g., Jordan, 1991; Bader and Weilenmann, 1992; Brun et al., 1989, 1992; Bartelt and Lehning, 2002; Sauter et al., 2020). While this is usually true for their mathemat-

ical formulation, any numerical implementation might cause violations of mass and energy conservation. Commonly, these numerical details do not receive sufficient attention. This is e.g. reflected by the fact that *none* of the previous snow or firn intercomparison projects (Slater et al., 2001; Etchevers et al., 2004; Essery et al., 2009; Krinner et al., 2018) contains an assessment of the accuracy of mass and energy conservation in the numerical implementation. This contrasts with dedicated numerical experiments conducted in other intercomparison projects, for example in the climate modelling community (Irv-

ing et al., 2021). Numerical errors are frequently argued to be small. But since snowpack model runs must cover seasons or centuries, even small numerical inconsistencies can cause drifts or lead to significant problems on these time scales. As we will show in this paper, achieving strict mass and energy conservation is particularly subtle in the highly coupled, non-linear situations outlined above.

In this study, we aim at pinpointing the most suitable numerical treatments based on criteria of time step requirements,

conservation of energy, and conservation of mass. To this end, we unify previous model developments within a comprehensive, stand-alone, finite-element core written in python. Contrary to Schürholt et al. (2022), each step of the numerical formulation and solution is coded internally, except for the inversion of the linearized system of equations which relies on the standard numpy linear algebra library. In this way, we have a complete control on every detail of the numerical recipe, and can thus explore various solving strategies. We demonstrate the improvements within established benchmark scenarios and carve out

the origin of errors in the numerical solution of the conservation laws. We also discuss the treatment of boundary conditions (BCs) on vapor.

The paper is organised as follows. In Sect. 2, we introduce the mathematical models derived by Calonne et al. (2014) and Hansen and Foslien (2015), and underline specific issues. In Sect. 3, we go through the details of our numerical implementation, specifying main differences compared to previous work. In Sect. 4, our model is tested on numerical benchmarks and





appropriate numerical approaches are highlighted. Sect. 5 summarises our work and is a an opening on implications of our findings for future work.

## 2 Mathematical models

In this section, we present the mathematical models derived by Calonne et al. (2014) and Hansen and Foslien (2015), and point out relevant issues.

### 2.1 The Calonne model

Starting from the physical phenomena occurring at the pore scale - specifically (i) the heat conduction through air and ice, (ii) the water vapor diffusion in the pore space, and (iii) the sublimation and deposition of vapor at the ice-pore interface - Calonne et al. (2014) used the two-scale expansion method in order to derive a closed system of equations governing the heat and water vapor budget at the macroscopic scale. The main advantage of this approach is that the exact expression of the

effective properties (such as the snow thermal conductivity) and of the source terms naturally arises as the macroscopic system of equations is derived. Yet, it must be stressed that Calonne et al. (2014) do not include any equation governing the evolution of the pore space at the micro-scale related to water molecules depositing on or sublimating from the ice-pore interface. As a consequence, this macroscopic model implicitly assumes an invariant microstructure, i.e. the ice volume fraction does not evolve over time. In addition, the two-scale expansions method is unsuited for high reaction rates (Bourbatache et al., 2021),

that is to say when vapor deposits easily on the ice. In other words, the domain of applicability of the macroscopic model proposed by Calonne et al. (2014) is bounded to cases for which the crystal growth velocity due to deposition (or sublimation) of water vapor molecules at the ice-pore interface is limited by the characteristic time of reaction rather than by the diffusion one (i.e. a low Damköhler number; Bourbatache et al., 2021). Under these two assumptions, the macroscopic system of equations writes:

$$
\begin{cases}
(\rho C_{\mathrm{p}})^{\mathrm{eff}}\, \partial_t T - \nabla \cdot \left( k^{\mathrm{eff}} \nabla T \right) = L_{\mathrm{m}} c \\
(1 - \Phi_{\mathrm{i}}) \partial_t \rho_{\mathrm{v}} - \nabla \cdot \left( D^{\mathrm{eff}} \nabla \rho_{\mathrm{v}} \right) = -c
\end{cases}, \tag{1}
$$

where $T$ is the temperature, $\rho_{\mathrm{v}}$ the water vapor density, $\Phi_{\mathrm{i}}$ the ice volume fraction, $L_m$ the specific latent heat of sublimation, $(\rho C_{\mathrm{p}})^{\mathrm{eff}}$ the effective heat capacity, $k^{\mathrm{eff}}$ the effective heat conductivity, and $D^{\mathrm{eff}}$ the effective vapor diffusion coefficient. The deposition rate $c$ is given by

$$
c = s\alpha v_{\mathrm{kin}}(\rho_{\mathrm{v}} - \rho_{\mathrm{v}}^{\mathrm{eq}}), \tag{2}
$$

with $s$ the surface area density per unit volume, which is assumed to be constant consistently with the implicit invariant-microstructure assumption, $v_{\mathrm{kin}} = \sqrt{(k_{\mathrm{B}} T)/(2\pi m_{H2O})}$ the kinetic velocity related to the velocity of water molecules in the pore space ($k_{\mathrm{B}}$ being Boltzmann's constant and $m_{H2O}$ the mass of a water molecule), $\rho_{\mathrm{v}}^{\mathrm{eq}}$ the saturation vapor density, and $\alpha$ the sticking coefficient of water molecules on the ice surface. Referring back to the considerations raised above regarding the





domain of applicability of this macroscopic model, it appears that the latter is valid provided that $\alpha \approx 10^{-3}$ or less (Fourteau
et al., 2021).

The saturation water vapor density $\rho_{\mathrm{v}}^{\mathrm{eq}}$ being a non-linear function of $T$, the macroscopic model proposed by Calonne et al.
(2014) amounts to a system of two two-way coupled non-linear diffusion-reaction equations which must be solved for $T$ and
$\rho_{\mathrm{v}}$. Because we use parameterizations that neglect the dependency of the effective parameters to $T$ (Appendix A), the only
non-linearity of the system arises from the source terms through $\rho_{\mathrm{v}}^{\mathrm{eq}}$ and $v_{\mathrm{kin}}$. In what follows, we refer to this model as the
Calonne model.

## 2.2    The Hansen model

Using mixture theory, Hansen and Foslien (2015) derived another macroscopic heat and water vapor conservation model.
Contrary to Calonne et al. (2014), they assumed the deposition/sublimation of water vapor to be fast enough so that small over-
saturations (under-saturations) in the pore space are corrected almost instantaneously by the deposition (sublimation) of water
molecules. Mathematically, such a situation arises when the product $\alpha v_{\mathrm{kin}}$ becomes very large. In this case, the deposition
rate corresponds to the rate at which this deposition (sublimation) of molecules must occur so that the water vapor density is
permanently and instantaneously restored to its saturated value, i.e. so that $\rho_{\mathrm{v}}(T) = \rho_{\mathrm{v}}^{\mathrm{eq}}(T)$ at any time. As for the Calonne
model, the equations of Hansen and Foslien (2015) are derived at constant microstructure. Based on these two assumptions,
the macroscopic energy conservation and water vapor mass balance can be written, respectively:

$$
\quad \begin{cases} (\rho C_{\mathrm{p}})^{\mathrm{eff}} \, \partial_t T - \nabla \cdot \left( k^{\mathrm{eff}} \nabla T \right) = L_{\mathrm{m}} c \\ (1 - \Phi_{\mathrm{i}}) \frac{d\rho_{\mathrm{v}}^{\mathrm{eq}}}{dT} \partial_t T - \nabla \cdot \left( D^{\mathrm{eff}} \frac{d\rho_{\mathrm{v}}^{\mathrm{eq}}}{dT} \nabla T \right) = -c \end{cases} , \tag{3}
$$

with the underlying assumption that

$$
\rho_{\mathrm{v}} = \rho_{\mathrm{v}}^{\mathrm{eq}}. \tag{4}
$$

In Hansen and Foslien (2015), the effective parameters $k^{\mathrm{eff}}$ and $D^{\mathrm{eff}}$ are intuited from synthetic microstructures and are not
a direct by-product of the mixture theory. The system of Eqs. (3) can be casted into a single one, which then writes:

$$
\quad \left( (\rho C_{\mathrm{p}})^{\mathrm{eff}} + (1 - \Phi_{\mathrm{i}}) L_{\mathrm{m}} \frac{d\rho_{\mathrm{v}}^{\mathrm{eq}}}{dT} \right) \partial_t T - \nabla \cdot \left[ \left( D^{\mathrm{eff}} L_{\mathrm{m}} \frac{d\rho_{\mathrm{v}}^{\mathrm{eq}}}{dT} + k^{\mathrm{eff}} \right) \nabla T \right] = 0. \tag{5}
$$

In the end, the model of Hansen and Foslien (2015) consists in a single prognostic equation on $T$ which has the form of a
non-linear diffusion equation. The non-linearity comes from the dependence to temperature of the derivative of the saturation
water vapor density that appears in the apparent heat capacity and apparent thermal conductivity. The deposition/sublimation
rate $c$ can then be diagnosed from one of the Eqs (3). In what follows, we refer to this model as the Hansen model.

## 2.3    General form of the equations governing the heat and vapor budgets in evolving pore space

Despite being derived under different assumptions, the Calonne and Hansen models comprise similarities. In both of these
works, the total energy budget accounts for the contribution of vapor transport and of the latent heat that is released or absorbed





whenever water vapor deposits or sublimates, which affects the water vapor mass balance in return. As noted by Schürholt et al. (2022), if the differences regarding the parameterizations of the effective parameters are put aside, the system of Eqs (3) can

be derived from the system of Eqs (1) under the assumption that $\rho_{\mathrm{v}} = \rho_{\mathrm{v}}^{\mathrm{eq}}$. Fundamentally, both systems of equations consist of an equation for the conservation of heat including a source term proportional to the deposition rate and an equation for the conservation of vapor including the same deposition rate as a sink term. However, the systems of equations differ in how they are closed: the Calonne model is closed by computing the deposition rate $c$ as a first order reaction, and the Hansen model is closed by assuming water vapor saturation.

Both models have been derived assuming an invariant microstructure. This restriction can be lifted by realising that the first terms of both the heat and vapor conservation equations correspond to the time derivatives of the total heat and total vapor content, respectively. These terms can then simply be rewritten to include the effect of an evolving ice volume fraction on the temporal evolution of these two quantities. A more subtle issue lies on the fact that both works neglected the settlement of the snowpack, and the resulting advection of material quantities. This problem is usually circumvented by solving the settlement

and heat/vapor budget problems separately. This corresponds to the Eulerian-Lagrangian framework described by Simson et al. (2021): (i) the settlement of the snowpack is inferred adopting a Lagrangian point of view, (ii) the computed settlement is used to deform the numerical mesh, and (iii) the heat and vapor equations are solved on the obtained mesh using an Eulerian approach. With this procedure, the contribution of advection in the evolution of the temperature, vapor and deposition rate fields is accounted for during the mesh deformation step, and thus vanishes from the heat/vapor conservation equations. In

view of all these considerations, the general form of the equations governing the heat, vapor, and ice budgets in snow in an Eulerian framework can be written:

$$
\begin{cases}
\partial_t h_{ice} - \nabla \cdot \left( k^{\mathrm{eff}} \nabla T \right) = L_m c \\
\partial_t \left[ (1 - \Phi_{\mathrm{i}}) \rho_{\mathrm{v}} \right] - \nabla \cdot \left( D^{\mathrm{eff}} \nabla \rho_{\mathrm{v}} \right) = -c \quad , \\
\partial_t \Phi_{\mathrm{i}} = \frac{c}{\rho_{\mathrm{i}}}
\end{cases}
\tag{6}
$$

where $h_{ice}$ is the heat content of snow and $\rho_{\mathrm{i}}$ the density of ice. We stress that, contrary to Eqs. (1) or (3), the accumulation term of the heat equation is written in terms of heat content and not in terms of temperature. This heat content is related to

temperature through:

$$
\frac{\partial h_{ice}}{\partial T} = (\rho C_{\mathrm{p}})^{\mathrm{eff}} .
\tag{7}
$$

One might be tempted to express $\partial_t h_{ice}$ as $(\rho C_{\mathrm{p}})^{\mathrm{eff}} \partial_t T$ by means of the chain rule. As thoroughly explained in, e.g., Tubini et al. (2021), while this is valid in the continuous partial differential equation (PDE), this is not necessarily the case for the

discrete domain. Specifically, application of the chain rule if $(\rho C_{\mathrm{p}})^{\mathrm{eff}}$ is not constant during the discrete solution of the equation will result in the non-conservation of energy (e.g. Celia et al., 1990; Casulli and Zanolli, 2010; Tubini et al., 2021). Therefore, we choose to explicitly keep the heat content $h_{ice}$ in the heat budget, and thus to express this equation in its so-called "mixed-form" (similarly to Celia et al., 1990).





The system of Eqs (6)-(7) contains five unknowns ($h_{ice}$, $T$, $\rho_{\mathrm{v}}$, $\Phi_{\mathrm{i}}$, and $c$) and can be closed either with Eq. (2) from the
Calonne model, or with Eq. (4) from the Hansen model. We stress that this system of equations alone is not sufficient to
compute the evolution of the ice volume fraction. The contribution of mechanical settlement of snow on the latter must also be
accounted for (Simson et al., 2021). Although this process could be described in an Eulerian framework through the use of an
advection term (i.e. $\partial_t \Phi_{\mathrm{i}} + \nabla \cdot (\mathbf{v}\Phi_{\mathrm{i}}) = 0$, where $\mathbf{v}$ is the settling velocity), as mentioned above, adopting a Lagrangian point
of view to describe the deformation of the snowpack provides a more natural framework to compute the inherent ice volume
fraction increase. In the absence of phase change, the evolution of $\Phi_{\mathrm{i}}$ is only due to settlement. If additionally, the dependence
of the settling rate on temperature through viscosity is neglected, then the equations governing heat and vapor on the one hand,
and the equations governing ice volume fraction on the other hand, become partly independent and can be solved sequentially.

Finally, the compaction of snow leads to a reduction of the pore space. Since, the air is free to escape during the settlement
of the snowpack, this decrease in porosity results in a net loss of dry air and vapor. If the effective heat capacity of snow
$(\rho C_{\mathrm{p}})^{\mathrm{eff}}$ is computed taking into account the heat carried by dry air, then this loss of dry air leads to a net loss of energy. While
physically relevant, this loss of energy though air ejection can be overlooked by considering that only the ice phase carries
energy in snow (Appendix A3). This assumption is justified by the small volumetric heat capacity of dry air compared to that
of ice. In contrast, the net loss of vapor is fully part of the considered problem, and should therefore be kept in mind when
closing the vapor budget from one time step to the next.

## 3 Numerical implementation

In this Section, we go through the most important features of our numerical implementation, and underline the main differences
compared to the published approaches of Simson et al. (2021) and Schürholt et al. (2022). All variables, parameters, and
constants used in our model are summarised in Table 1.

### 3.1 Numerical strategy

**Spatial and temporal discretization**

The spatial discretization is based on the FEM. Indeed, the FEM enables a clear distinction between element-wise and nodal
variables. The former are constant over elements and typically correspond to conserved quantities (e.g., ice volume fraction),
while the latter are defined at mesh nodes, are continuous in space, and typically correspond to physical quantities driving the
fluxes in between elements (e.g., temperature). As explained in further details in Sect. 3.3, such a vision is consistent with
the fact that the conservation of energy and mass are fulfilled on average per spatial intervals rather than locally. In contrast
to the work of Schürholt et al. (2022) who used a python-based FEM platform, we are coding internally every steps of the
method except the inversion of the linear system. This allows us to have complete control over the numerical implementation.
On the other hand, this enables to design a code structure well-suited for snowpack modelling, in which each physical process
corresponds to a module that can be easily activated or de-activated at user convenience. Below, we briefly recall the basics of





**Table 1.** List of variables, parameters and constants used in our model.

| Symbol | Name | Equation/Value | Unit |
|---|---|---|---|
| Nodal variables | | | |
| $c$ | Deposition rate | Eqs. (2), (3) | kg m$^{-3}$ s$^{-1}$ |
| $H$ | Enthalpy content | Eq. (15) | J m$^{-3}$ |
| $T$ | Temperature | Eqs. (1), (5), (15) | K |
| $\rho_\mathrm{v}$ | Vapor density | Eqs. (1), (4) | kg m$^{-3}$ |
| $\sigma$ | Stress | Eq. (18) | Pa |
| Element-wise variables | | | |
| $E$ | Energy per element | Eq. (14) | J m$^{-2}$ |
| $\Phi_\mathrm{i}$ | Ice volume fraction | Eq. (21) | – |
| Parameters | | | |
| $D^\mathrm{eff}$ | Effective vapor diffusion coefficient | Eqs. (A1), (A2) | m$^2$ s$^{-1}$ |
| $k^\mathrm{eff}$ | Effective heat conductivity | Eqs. (A3), (A4) | W m$^{-1}$ K$^{-1}$ |
| $v_\mathrm{kin}$ | Kinetic velocity | Eq. (2) | m s$^{-1}$ |
| $s$ | Surface area density per unit volume | 3770 | m$^{-1}$ |
| $\alpha$ | Sticking coefficient (value by default) | $5 \times 10^{-3}$ | – |
| $\eta$ | Effective viscosity | Eq. (A7) | Pa s$^{-1}$ |
| $(\rho C_\mathrm{p})^\mathrm{eff}$ | Effective heat capacity | Eq. (A5) | J m$^{-3}$ K$^{-1}$ |
| $\rho_v^{eq}$ | Saturation water vapor density | Eq. (A6) | kg m$^{-3}$ |
| Constants | | | |
| $C_\mathrm{a}$ | Air heat capacity | 1005 | J m$^{-3}$ K$^{-1}$ |
| $C_\mathrm{i}$ | Ice heat capacity | 2000 | J m$^{-3}$ K$^{-1}$ |
| $D_0$ | Vapor diffusion coefficient in air | $2 \times 10^{-5}$ | m$^2$ s$^{-1}$ |
| $g$ | Gravitational constant | 9.81 | m s$^{-2}$ |
| $k_\mathrm{a}$ | Air heat conductivity | 0.024 | W m$^{-1}$ K$^{-1}$ |
| $k_\mathrm{i}$ | Ice heat conductivity | 2.3 | W m$^{-1}$ K$^{-1}$ |
| $k_\mathrm{B}$ | Boltzmann constant | $1.38 \times 10^{-23}$ | J K$^{-1}$ |
| $L_m$ | Specific latent heat of sublimation | 2835333 | J kg$^{-1}$ |
| $m_{H20}$ | Mass of a water molecule | $2.991507 \times 10^{-26}$ | kg |
| $\rho_\mathrm{i}$ | Ice density | 917 | kg m$^{-3}$ |

the FEM. For more details, we refer the reader to one of the many books that have been written on the subject (e.g., Pepper and Heinrich, 2005).





The exact (analytical) solution $u$ of a PDE being usually out of reach, the idea of the FEM is to find an approached (numerical) solution $\tilde{u}$ under the form:

$$\tilde{u}(\mathbf{x},t) = \sum_{j=1}^{N_{DOF}} \varphi_j(\mathbf{x})u_j(t) \quad \forall \mathbf{x} \in \Gamma, \tag{8}$$

where $\Gamma$ is the considered domain, and the $u_j$ are the discrete unknown scalar values of the problem to be solved. The functions $\varphi_j$ are linearly independent functions of space, referred to as the shape functions. Here, we follow the classical approach and adopt first-order Lagrangian polynomials as shape functions: the shape function $\varphi_j$ is a continuous piece-wise linear function, whose value equals 1 at the $j^{\text{th}}$ node and 0 at all other nodes. In this case, the unknown scalar value $u_j$ simply corresponds to the value of $\tilde{u}$ at the $j^{\text{th}}$ node, and the shape functions $\varphi_j$ can be viewed as linear interpolators in between nodes. The restriction

of shape functions to first-order polynomials is motivated by the results of Schürholt et al. (2022) that have reported that, in their experiments, increasing the polynomial order was equivalent to increasing the mesh resolution with the corresponding number of nodes.

The PDE is then rewritten in its weak form, which consists in multiplying the latter by any arbitrary test function, and to integrate this product over $\Gamma$. Integration by parts is carried out on the divergence term to weaken the differentiation requirement

on the solution field, which naturally makes boundary normal flux integrals arise. In order to obtain a closed set of discrete equations, the arbitrary test function is replaced by a finite set of test functions $\varphi_i$ that are taken equal to the shape functions (standard Galerkin procedure). The problem is then reduced to solving $N_{DOF}$ algebraic equations that can be casted into matrix forms, with matrix terms defined as integrals over space (see Appendix B). These integrals are evaluated using Gaussian quadratures. The latter consists in replacing a continuous integral by a weighted sum of function values at specific points, the

so-called Gaussian points, located within the elements. In particular, model parameters that depend on model variables are evaluated directly at Gaussian points at which nodal variables are interpolated through shape functions. When the considered model parameter is a non-linear function of some model variable, this is different from evaluating the model parameter at nodes and then using shape functions to interpolate the obtained values at Gaussian point as done in, e.g., Schürholt et al. (2022). We use a default value of two Gaussian points per element, but this setting could easily be changed.

While the FEM provides a spatial discretization scheme, the obtained matrix equation also needs to be discretized in time. The theta method is the most commonly adopted. It consists in expressing all quantities as a weighted average between their values at previous time step and their values at current time step. In general, the matrix form of the resulting algebraic system writes:

$$\underbrace{\left[\mathbf{M} + \theta\Delta t\mathbf{K}^{n+1}\right]}_{\text{Left-Hand Side Matrix } \mathbf{A}} \mathbf{U}^{n+1} = \underbrace{\left[\mathbf{M} + (\theta-1)\Delta t\mathbf{K}^n\right]\mathbf{U}^n + \Delta t\left[\theta\mathbf{F}^{n+1} + (1-\theta)\mathbf{F}^n\right]}_{\text{Right-Hand Side Vector } \mathbf{B}} + \Delta t\mathbf{F}^{\partial\Gamma}, \tag{9}$$

where the superscript $n+1$ (resp. $n$) is applied to quantities evaluated at current (resp. previous) time step, $\Delta t$ is the time step size, and $\mathbf{U}$ the solution vector. The matrices $\mathbf{M}$ and $\mathbf{K}$ are referred to as, respectively, the mass and stiffness matrices. The vector $\mathbf{F}$ is called the force vector, and the vector $\mathbf{F}^{\partial\Gamma}$ corresponds to boundary fluxes entering/leaving the domain during the time step. The particular cases $\theta = 0$, $\theta = 0.5$ and $\theta = 1$ correspond to, respectively, the first-order explicit Euler, the





Crank-Nicholson, and the first-order implicit Euler methods. It can be shown that whenever $0.5 \leq \theta \leq 1$ the time-stepping
algorithm is L2-stable: the error between the continuous time-derivative and its discrete counter-part remains bounded without
any requirement on the time step size. However, if the considered PDE is non-linear, the system (9) also becomes non-linear
for any $\theta > 0$. Such a system must be linearized. Linearization algorithms, such as Picard or Newton, are iterative methods that
require the prescription of an initial guess, which must be sufficiently close to the solution to ensure convergence of the algo-
rithm. The fulfilment of this condition hampers the prescription of arbitrary large time step, even when the time discretization
method is said to be unconditionally stable. On the other hand, while every cases with $\theta \neq 0.5$ are first-order in time (i.e., the
discretization error decreases linearly with $\Delta t$, as $\Delta t$ converge to zero), the Cranck-Nicholson method is second-order in time
(i.e., the discretization error decreases quadratically with $\Delta t$, as $\Delta t$ converge to zero). This higher accuracy of the Cranck-
Nicholson combined to its stability are often invoked to justify its use (e.g., Bader and Weilenmann, 1992; Decharme et al.,
2011; Schürholt et al., 2022). Yet, the obtained solution can be affected by spurious (decaying) oscillations if the time step is
too large or the element size too small. For this reason, we prefer to use the standard first-order implicit Euler method ($\theta = 1$).
Although potentially less accurate than the Cranck-Nicholson method, the latter is more stable and less prone to oscillations
(Formaggia and Scotti, 2011).

Specific BCs can be implemented through appropriate modification of the matrix system (9). More specifically, Neumann
and Robin BCs can be directly implemented through the boundary flux vector $\mathbf{F}^{\partial \Gamma}$. Dirichlet BC at a boundary node can be
implemented by replacing its corresponding line in the left-hand side (l.h.s) matrix $\mathbf{A}$ of Eq. (9) with the line of the identity
matrix and setting the Dirichlet value in the right-hand side (r.h.s) vector $\mathbf{B}$.

Once the solution vector has been obtained, it is possible to diagnostic the fluxes entering/leaving the system by calculating
the matrix system residual, defined as $\Delta t \mathbf{F}^{\partial \Gamma} = \mathbf{A} \mathbf{U} - \mathbf{B}$. Note that this operation applies for all types of BCs, notably Dirichlet
conditions for which the boundary fluxes are not explicitly prescribed but nonetheless exist to maintain the boundary solution at
its prescribed value. Assessing the boundary fluxes is necessary to verify the closure of the energy budget, and can be required
for latter coupling to external models.

**Coupling scheme**

One of the main goal of our work is to highlight the importance of a coupled numerical solution of the heat and vapor transport
processes. Details on the various approaches that have been implemented and compared to come up to this conclusion are given
in Sect. 3.2. In contrast, the ice mass conservation is systematically solved in a separated step following the same first-order
operator splitting approach as Simson et al. (2021) and Schürholt et al. (2022). Yet, as demonstrated in Sect. 2, heat/vapor
transfer and deposition are two-way coupled: a perturbation of the $T$ and/or $\rho_v$ fields affects the deposition rate field, which in
turns changes the distribution of $\Phi_i$, which echoes on the distributions of the $\Phi_i$-dependent parameters $(\rho C_p)^{\text{eff}}$, $k^{\text{eff}}$ and $D^{\text{eff}}$,
which impacts the fields of $T$ and $\rho_v$, and so on. Therefore, one has to be aware that solving these two processes sequentially
will necessarily introduce some error. Specifically, as the energy and vapor mass budget are solved assuming a constant mi-





crostructure, the consecutive modification of the latter through deposition/sublimation breaks the previously computed energy and vapor mass budgets. This results in non-physical energy and vapor mass sources/sinks, that we referred to as energy/mass

leakage. However, in most of the natural configurations, the variation of the ice mass related to deposition within targeted time steps of the order of $15\ \mathrm{min}$ is expected to remain negligible compared to the total ice mass of the simulated domain. It follows that the energy leakage remains limited, and its associated error in the global energy budget is well identified. Note also that, because our settlement scheme is designed to conserve the ice mass perfectly in the absence of phase change (see Sect. 3.3), there is no such problem of energy leakage for what regards the settlement-induced evolution of $\Phi_i$. For all these reasons, we

consider that the operator-splitting approach is acceptable for what regards the ice mass conservation equation.

Figure 1a summarises the general structure of the model. Within one time step, we solve first the heat and vapor transfer process, which can be modelled either through the system of Calonne or through the one of Hansen (Fig. 1b). Independently of the considered system, this is done with the distribution of $\Phi_i$ from the previous time step. Next, the nodal field of stress

is updated from the weight contained in all overlying elements, the latter being calculated from the distribution of $\Phi_i$ from the previous time step. Then, the settlement solver is executed in order to consistently update the mesh node positions and the element-wise field of $\Phi_i$. Finally, a solver is executed to diagnose the amount of energy contained in the domain from the nodal fields of $T$ and $\rho_v$ (for the system of Calonne and the system of Hansen in its T-form, i.e. when using $T$ as the prognostic variable) or from the nodal field of enthalpy (for the system of Hansen in its mixed form). At the very end of the time step, the

global ice mass and energy budgets are evaluated to check for conservation. Note that any of the solvers can be activated or de-activated depending on the processes of interest. In particular, the settlement-related and deposition-related evolution of $\Phi_i$ can easily and independently be switched on or off.

**Computational domain and notation**

As illustrated in Fig. 2, the snowpack is vertically discretized on a one dimensional finite-element grid. The $z$ axis is oriented

upward, with $z = 0$ corresponding to the soil/snow interface. The initial position of nodes are based on the user-prescribed initial snowpack height so that the mesh is initially uniform. In the presence of mechanical settlement, the mesh will deform non-uniformly. Note that the application of the FEM to non-uniform mesh is straightforward. In this work, the problem of remeshing has not been investigated but is discussed in Sect. 5. We distinguish between variables defined at nodes and variables defined element-wise (Table 1). In what follows, the former are identified with the subscripts $k$ which corresponds to the node

number, while the latter are identified with the subscripts $k + 1/2$ which corresponds to the element number. The element $k + 1/2$ is comprised between nodes $k$ and $k + 1$. The total number of nodes is denoted $N_z$.



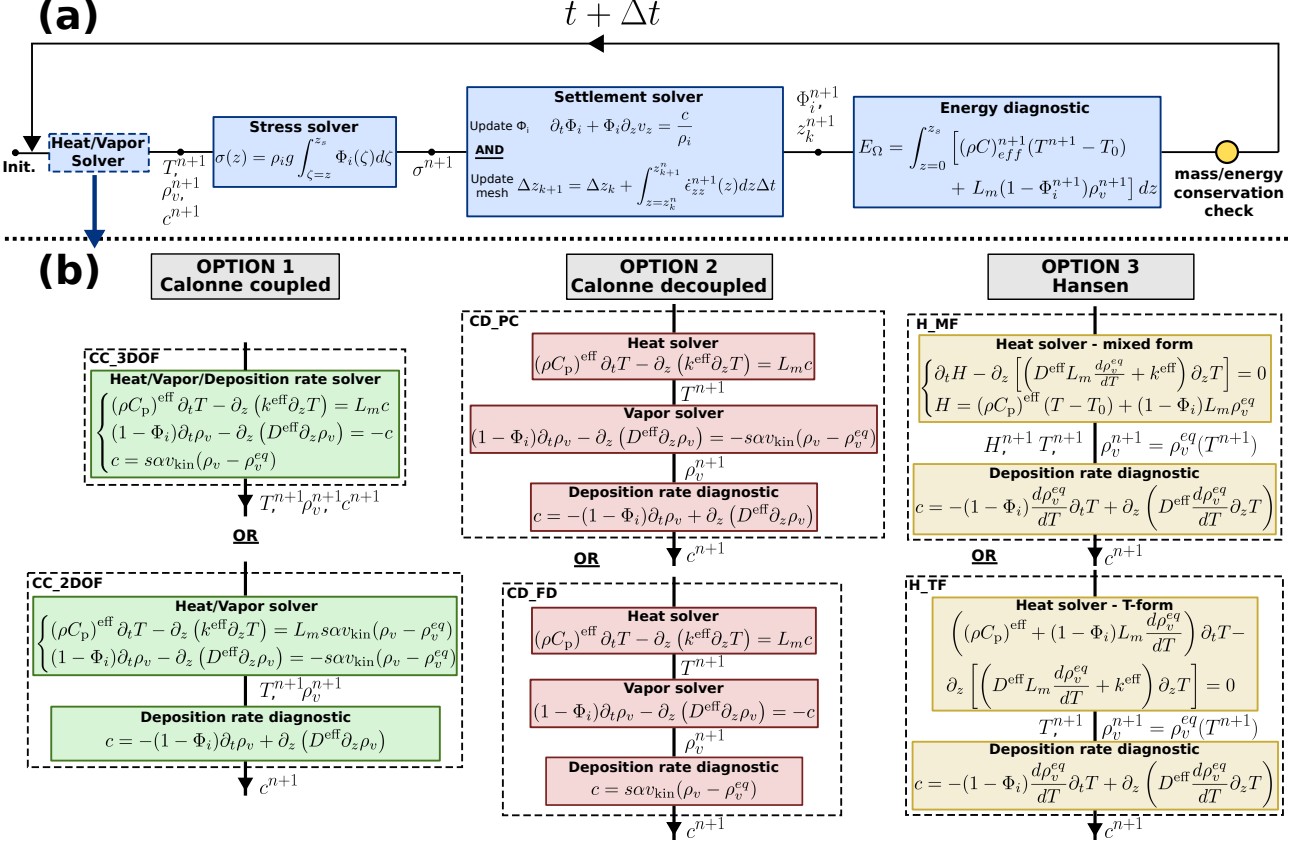

**Figure 1.** Panel **(a)** shows the general structure of the model. Panel **(b)** illustrates the three approaches implemented to compute the fields of temperature, water vapor and deposition rate: the coupled solution of the Calonne system, the decoupled solution of the Calonne system, and the solution of the Hansen system. For each of these approaches, two strategies have been considered that are described in the text and Appendix B.

## 3.2 Numerical solution of heat and water vapor transfer

**Resolution of the system of Calonne**

As illustrated in Fig. 1b, there are two possible approaches to solve the system of Eqs (1)-(2): either the whole system can be
solved in a coupled way (green boxes in Fig. 1b); or the three equations can be solved sequentially (red boxes in Fig. 1b). Two strategies are investigated for the coupled approach. The first strategy, denoted CC_3DOF, consists in solving the whole system (1)-(2) at once for the solution vector $\mathbf{u} = (T, \rho_v, c)$. This strategy corresponds to the one adopted by Schürholt et al. (2022) to treat the Calonne system. The second strategy, denoted CC_2DOF, consists in injecting Eq. (2) in the r.h.s terms of Eqs. (1), and solve the latter system for the solution vector $\mathbf{u} = (T, \rho_v)$. The $c$-field is then diagnosed from the obtained $\rho_v$-field
through the water vapor mass balance equation. Two strategies are also considered for the sequential treatment of the system.




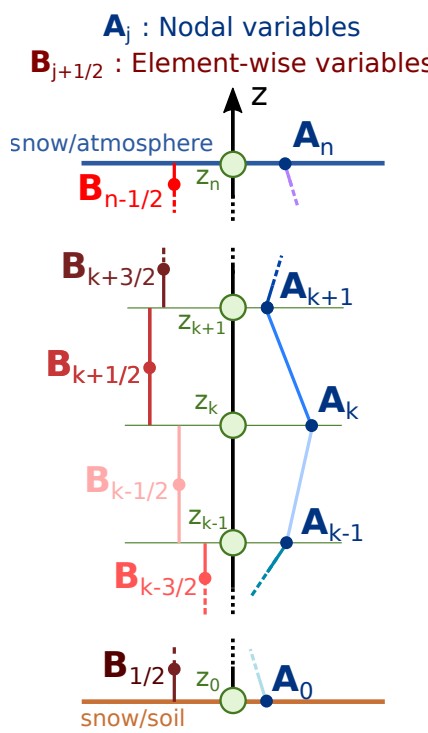

**Figure 2.** Illustration of the computational domain. The nature of the various model variables is summarised in Tab. 1.

In the strategy denoted CD_PC, the heat equation is solved first with a source term that is fixed from the $c$-field computed at the previous time step. Then, the water vapor mass balance is solved under its diffusion-reaction form: the r.h.s of the equation is replaced by its form (2), which introduces a reaction term on $\rho_v$ while the previously evaluated $T$-field is used to compute the source term stricto sensu, i.e. $s\alpha v_{\mathrm{kin}}(T)\rho_v^{\mathrm{eq}}(T)$. Finally, the $c$-field is updated as the closure of the water vapor mass balance

from the computed $\rho_v$-field. The second strategy, denoted CD_FD, is similar to the CD_PC strategy, except that both the heat and water vapor mass balance equations have their r.h.s fixed from the $c$-field computed at the previous time step. The latter is updated in a third step from Eq. (2) using the obtained $T$ and $\rho_v$-fields. The resulting matrix systems are summarised in Appendix B.

In order to manipulate standard mass matrices in which all non-zero terms correspond to the product of test and shape func-

tions, FEM softwares are prone to deal with a particular form of the heat equation in which the thermal diffusivity $k^{\mathrm{eff}}/(\rho C_{\mathrm{p}})^{\mathrm{eff}}$ is assigned to the flux divergence operator, rather than keeping the heat capacity $(\rho C_{\mathrm{p}})^{\mathrm{eff}}$ as a factor of the accumulation term. Similarly, it can be tempting to divide the water vapor mass balance equation by the factor $(1 - \Phi_{\mathrm{i}})$ assigned to the accumulation term in order to deal with the generic form of a diffusion-reaction equation. These operations are performed in, e.g., the FEniCS code as published by Schürholt et al. (2022). Practically, this corresponds to assigning the inverse of these factors

to the stiffness matrix and force vector rather than keeping them in the mass matrix. However, as soon as these factors are non-uniform, which is the case for snow in general as $\Phi_{\mathrm{i}} = f(z)$ and $(\rho C_{\mathrm{p}})^{\mathrm{eff}} = f(\Phi_{\mathrm{i}})$, the conservative forms of Eqs (1) are



not equivalent to these reformulations. Therefore, in order to preserve the conservative properties (at given microstructure) of the system of Calonne in the discrete domain, we assign the factors $(\rho C_{\mathrm{p}})^{\mathrm{eff}}$ and $(1 - \Phi_{\mathrm{i}})$ directly to the mass matrix (Appendix B). In the following, we refer to this form of the mass matrix as the proper form of the mass matrix.

By nature, the system of Eqs. (1) respect the maximum principle in the continuous domain. From a physical perspective, the maximum principle states that, in the absence of phase change, the maximum value of $T$ (resp. $\rho_{\mathrm{v}}$) is reached either at initial time or at the boundaries (Protter and Weinberger, 2012). In order to get a uniform convergence of the numerical solution to the exact one and avoid non-physical extrema in the interior of the domain, it can be shown that a discrete counterpart of this principle, the so-called discrete maximum principle (DMP), must be fulfilled (Ciarlet, 1973). Yet, in some situations the
FEM is inclined to violate the DMP due to, among others, the treatment of time derivatives (e.g., Celia et al., 1990), and/or of reaction terms (e.g., John and Schmeyer, 2008). A sufficient condition for the DMP to be respected is that the matrix $\mathbf{A}$ of the system (9) has the following properties: (i) all diagonal terms are positive, (ii) all off-diagonal terms are negative or zero, and (iii) the row sums are positive (John and Schmeyer, 2008). Thus, a commonly-used method to enforce the matrix system to satisfy the DMP without adding further constraints on the time step is to lump the mass matrix and/or the reactive part of
the stiffness matrix. This operation consists in replacing the diagonal term of each row of the considered matrix by the sum of all terms of the row, while all off-diagonal terms are forced to zero (e.g., Milly, 1985; Celia et al., 1990; John and Schmeyer, 2008; Thomée, 2015). In the present work, this method is adopted whenever spurious oscillations show up. As we will show, although the obtained solution fields are very slightly modified, the method enables to remove spurious oscillations without affecting energy conservation.

As mentioned in Sect. 2.1, the non-linearity of the system of Calonne is related to the dependence of the source terms, through parameters $\rho_{\mathrm{v}}^{\mathrm{eq}}$ and $v_{\mathrm{kin}}$, to temperature (Appendix A4). As a consequence, this non-linearity vanishes whenever these parameters are computed from a $T$-field obtained in a separated step. This is the case for both strategies based on the decoupled solution of the system. In contrast, as soon as a coupled solution of the system is considered, a linearization procedure is required. For both strategies of the coupled approach, we implement a linearization algorithm which mixes a Picard method
for $v_{\mathrm{kin}}$, and a Newton method for $\rho_{\mathrm{v}}^{\mathrm{eq}}$. Practically, when solving the system (9) at iteration $k+1$ for iterate $\mathbf{u}^{k+1}$, the value of $v_{\mathrm{kin}}^{k+1}$ is fixed using the temperature field obtained at previous iteration, i.e. $v_{\mathrm{kin}}^{k+1} = v_{\mathrm{kin}}(T^k)$. Within the same iteration step, the value of $\rho_{\mathrm{v}}^{eq,k+1}$ is evaluated as:

$$\rho_{\mathrm{v}}^{eq,k+1} = \left. \frac{d\rho_{\mathrm{v}}^{\mathrm{eq}}}{dT} \right|_{T=T^k} (T^{k+1} - T^k) + \rho_{\mathrm{v}}^{\mathrm{eq}}(T^k). \tag{10}$$

The first iterate corresponds to the solution vector obtained at the previous time step $T^n$. The convergence criterion of the
linearization algorithm reads:

$$2 \frac{\left| |\mathbf{u}^{k+1}| - |\mathbf{u}^k| \right|}{|\mathbf{u}^{k+1}| + |\mathbf{u}^k|} < \varepsilon, \tag{11}$$

where the tolerance criterion is set to $\varepsilon = 10^{-5}$ by default.

     Once the linearization algorithm has converged, the residuals of the system are evaluated to assess the sensible heat $\Delta t F_{\mathrm{T}}^{\partial\Gamma}$ and water vapor mass $\Delta t F_{\mathrm{P}}^{\partial\Gamma}$ that have entered or left the domain over the time step. The energy budget is evaluated at the





very end of the time step by comparing the evolution of the energy contained in the domain since the previous time step $\Delta E_\Omega$ to the aforementioned boundary fluxes. In practise, we compute:

$$E_{leak} = \Delta E_\Omega - \Delta t(F_{\mathrm{T}}^{\partial\Gamma} + L_{\mathrm{m}}F_{\mathrm{P}}^{\partial\Gamma}) - \Delta t L_{\mathrm{m}}F_{\mathrm{P,set.}}^{\partial\Gamma}, \tag{12}$$

where $E_{leak}$ is the energy leakage, which is zero if energy is conserved. As mentioned in Sect. 2, because the contribution of dry air in the effective heat capacity of snow is neglected, the dry air expelled from the snowpack in response to settlement does not affect the energy budget. On the contrary, the latent energy loss due to water vapor leaving the domain as the snowpack settles must be accounted for through the term $\Delta t L_{\mathrm{m}}F_{\mathrm{P,set.}}^{\partial\Gamma}$. The water vapor mass leaving the snowpack over $\Delta t$ is diagnosed in the settlement solver as:

$$\Delta t F_{\mathrm{P,set.}}^{\partial\Gamma} = \sum_{k=1}^{N_z-1} \left[ \int_{\zeta=z_k^n}^{z_{k+1}^n} \rho_v^{n+1} d\zeta - \int_{\zeta=z_k^{n+1}}^{z_{k+1}^{n+1}} \rho_v^{n+1} d\zeta \right]. \tag{13}$$

The total energy contained in the domain $E_\Omega$ is diagnosed as the sum of the energies contained in each element in the last executed solver (Fig. 1a). The energy contained in the element $k+1/2$ is calculated as:

$$E_{k+1/2} = \int_{\zeta=z_k^{n+1}}^{z_{k+1}^{n+1}} \left[ (\rho C)_{eff}^{n+1}(T^{n+1} - T_0) + L_{\mathrm{m}}(1 - \Phi_i^{n+1})\rho_v^{n+1} \right] d\zeta, \tag{14}$$

where $T_0$ is an arbitrary reference temperature, which we set to $T_0 = 273$ K.

**Resolution of the system of Hansen**

For the system of Hansen, also two implementation strategies are investigated (yellow boxes in Fig. 1b). In the first strategy, denoted H_MF, the prognostic equation on $T$ is treated under its mixed-form. Concretely, this consists in solving the following system for the solution vector $\mathbf{u} = (H, T)$:

$$\begin{cases} \partial_t H - \partial_z \left[ \left( D^{\mathrm{eff}} L_{\mathrm{m}} \frac{d\rho_v^{\mathrm{eq}}}{dT} + k^{\mathrm{eff}} \right) \partial_z T \right] = 0 \\ H = (\rho C_{\mathrm{p}})^{\mathrm{eff}} (T - T_0) + (1 - \Phi_i)L_{\mathrm{m}}\rho_v^{\mathrm{eq}} \end{cases} \tag{15}$$

where $H$ is the enthalpy content defined as a nodal variable. Because $H$ is a $\Phi_i$-dependent prognostic variable, it must imperatively be updated within the settlement solver after $\Phi_i$ has been updated. In order to illustrate the problem of violation of energy conservation when a chain rule is performed in the discrete domain (Sect. 2.2), we consider a second strategy, denoted H_TF, which consists in solving the prognostic equation of the Hansen system under its usual T-form (5). The resulting matrix systems are summarised in Appendix B. The H_TF strategy is the one adopted by, e.g., Simson et al. (2021) and Schürholt et al. (2022). Independently of the chosen strategy, the field of $\rho_v$ is deduced from the obtained $T$-field assuming that water vapor density is saturated everywhere. Then, the second Eq. of (3) is solved for the field of $c$.





For the T-form of the prognostic equation, we are careful to assign the apparent heat capacity directly to the mass matrix (proper form of the mass matrix). Note that ambiguities do not arise for the mixed-form of the equation as it does not include any multiplying factor in the accumulation term. Furthermore, as for the Calonne system, spurious oscillations related to the violation of the DMP can occur. Again, this difficulty is overcome by lumping the mass matrices of both the prognostic equation on $T$ (or $H$ for the mixed-form) and the diagnostic equation on $c$ when necessary. As mentioned in Sect. 2.2, both the mixed

and T-form of the prognostic equation are non-linear due to the dependence of $d\rho_v^{eq}/dT$ to $T$ (Appendix A4). This non-linearity is treated through the implementation of a Picard linearization loop. As for the coupled Calonne system, the tolerance criterion is set to $\varepsilon = 10^{-5}$ by default. Note that the approach of Simson et al. (2021), which consists in linearizing the equation by fixing the value of $d\rho_v^{eq}/dT$ from the $T$-field obtained at previous time step, is equivalent to perform a single Picard iteration. In this case, the obtained solution does not correspond to the implicit solution of the problem and thus does necessarily possess

its stability features.

    As for the Calonne system, the residuals are evaluated immediately after the convergence of the linearization algorithm. Note that, because the Hansen system contains only one prognostic equation that mixes up the contributions of latent and sensible heat to the energy budget, the obtained residuals correspond to the total enthalpy fluxes at boundaries, without any references as to how these fluxes are split between sensible and latent heat fluxes. Again, these boundary fluxes are used to close the

energy budget. The total energy contained in the domain is directly obtained through integration of the nodal enthalpy over the elements for the mixed-form case. For the T-form case, it is evaluated from $T$ based on Eq. (14) using the assumption of saturated water vapor density. As for the Calonne system, the energy budget accounts for energy loss related to the water vapor mass leaving the snowpack when settlement is activated through Eq. (13).

    Although the field $c$ is diagnosed a posteriori from the obtained $T$-field based on the second Eq. of (3), this solution is also

based on the FEM, and a boundary flux term naturally appears in the force vector through integration by parts of the divergence operator. In the absence of any prescription from the user, the natural BC applies and this term is forced to be zero. Yet, the existence of a boundary flux strongly affects the deposition rate at the corresponding boundary node, as the local over/under-saturation governs the magnitude of the deposition/sublimation that is necessary to maintain vapor saturation. It follows that the prescription of proper vapor fluxes at boundaries is an integral part of the physical problem and should ideally arise from

a vapor budget at interfaces, similarly to the standard surface energy budget. On the other hand, fixing $c$ to an arbitrary value at boundaries (Dirichlet type BCs), as done by Simson et al. (2021), implicitly implies the prescription of a water vapor mass flux that adjusts itself so that saturation is maintained with a fixed contribution of deposition/sublimation. This boundary vapor flux then corresponds to the residuals of the vapor mass budget equation. This topic is further explored in Sect. 4.6.

### 3.3 A mass-conservative Lagrangian settlement scheme

**General considerations**


Our settlement scheme is based on the method of characteristics as presented in Simson et al. (2021) with some corrections to guarantee ice mass conservation, as well as a more explicit formulation of the element-wise nature of this conservation.



The general idea of this approach is that the mesh nodes should move at the velocity of the ice matrix as the latter settles so that all ice mass fluxes related to settlement wipe out in the working frame, thus resorting to a Lagrangian coordinate system.

Concretely, this allows to eliminate the advection term $\mathbf{v} \cdot \nabla \Phi_\mathrm{i}$ from the continuity equation, which then writes in 1D:

$$\partial_t \Phi_\mathrm{i} + \Phi_\mathrm{i} \partial_z v_z = \frac{c}{\rho_\mathrm{i}}, \tag{16}$$

where $\partial_z v_z$ is the divergence of the settling velocity which, in 1D, directly corresponds to the vertical strain rate $\dot{\epsilon}_{zz}$. The vertical strain rate relates to the vertical stress $\sigma_{zz}$ through the constitutive law of snow, which we take as a simple linear viscous law:

$$\dot{\epsilon}_{zz} = \frac{1}{\eta} \sigma_{zz}, \tag{17}$$

where $\eta$ is the effective viscosity of snow and $\sigma_{zz}$ is the vertical stress. The effective viscosity is an important and poorly constrained snow property that depends, among others, on microstructure and temperature (e.g., Wiese and Schneebeli, 2017). Here, we use the viscosity parameterization of Vionnet et al. (2012) where the viscosity increases exponentially with density and decreases exponentially with temperature according to Eq. (A7). Note that Simson et al. (2021) have also considered the

case of a non-linear Glen's flow law, as well as the case of a constant viscosity. As the goal of this study is not an assessment of the model sensitivity to parameterization choices but to the details of the numerical treatment of the equations, we do not consider these cases here. Neglecting the contribution of air in the effective density of snow, the momentum conservation equation relates the distribution of the vertical stress to the one of the ice volume fraction as follows:

$$\partial_z \sigma_{zz} = \rho_\mathrm{i} g \Phi_\mathrm{i}(z), \tag{18}$$

where $g$ is gravity.

As underlined by Simson et al. (2021), Eq. (16) and its Lagrangian perspective corresponds to the strategy employed in all available detailed snowpack models to represent settlement, although it is usually not explicitly stated. Concretely, the constitutive law of snow is used to relate the stress supported by a layer, calculated as the cumulative weight of all overlying layers, to its total deformation. This total deformation is then used either to update directly the layer thickness defined as a state

variable (e.g., Jordan, 1991; Vionnet et al., 2012), or to update the mesh coordinates using the fact that the node at the soil/snow interface does not move (e.g., Bader and Weilenmann, 1992; Bartelt and Lehning, 2002). The effective density, defined as a layer property, is then simply updated so that, in the absence of phase change, the mass contained within each layer remain the same before and after settlement. The layer-based nature of this scheme is obvious, in that conservation is not fulfilled locally but on average over finite space intervals referred to as control volumes (e.g., Jordan, 1991), elements (e.g., this study,

Bartelt and Lehning, 2002), or layers (e.g., Bader and Weilenmann, 1992; Bartelt and Lehning, 2002; Vionnet et al., 2012), depending on the numerical method chosen to solve the other PDEs of the model. In contrast, as Eq. (16) expresses the local ice mass conservation, one may assume that its actual solution would enable to switch from this traditional layer-based vision to a more continuous description of the snowpack. We stress that this impression relies on a confusion between numerical and physical layers. Indeed, although all physical quantities involved in Eqs. (16)-(18) are continuous functions of $z$ and can





be calculated anywhere in the continuous domain, as soon as we go through the necessary numerical discretization step, the continuous vision breaks and we step back to a discrete description in which the ice phase is conserved on average over finite space intervals that can be seen as numerical layers. The settlement scheme then consists in two numerical operations that must be done in parallel and in a consistent way to ensure that conservative properties of Eq. (16) are preserved: (i) the update of the mesh node positions so that the variation of the mass contained in each element over one time step is entirely due to phase change and not at all to settlement, and (ii) the update of the ice volume fraction defined as an element-wise state variable accounting for the possible source term due to phase change.

**Update of mesh node positions**

The displacement $\Delta z_{k+1}$ of node $k+1$ between $t$ and $t+\Delta t$ corresponds to the displacement $\Delta z_k$ of the underlying node $k$ to which the total deformation of the space interval between $k$ and $k+1$ over $\Delta t$ must be added. This condition can be written:

$$\Delta z_{k+1} = \Delta z_k + \int_{z_k^n}^{z_{k+1}^n} \dot{\epsilon}_{zz}^{n+1}(z) dz \Delta t,$$
(19)

where $\dot{\epsilon}_{zz}^{n+1}$ is the vertical strain rate that, in 1D, directly corresponds to the divergence of the vertical velocity field $\partial_z v_z$. By exploiting the fact that the node located at the soil/snow interface is immobile, we can trace the displacement of all nodes of the domain through Eq. (19). A difficulty arises in the numerical treatment of the integral term of Eq. (19) after spatial discretization. Indeed, the numerical integration requires an assumption on the spatial variation of the integrand in between nodes. While this assumption is made a priori through the choice of the shape functions in the FEM, only the nodal values of the fields are considered to constitute the solution in the finite-difference method employed by Simson et al. (2021), without any explicit reference as to how should these fields vary in between nodes (Patankar, 1980). In their mesh deformation procedure, Simson et al. (2021) do not directly use the strain rate $\dot{\epsilon}_{zz}$ to evaluate the total deformation of the space intervals in between nodes, but perform a numerical integration to recover the settling velocity $v_z$ and use the latter to move the nodes. A careful inspection of Eq. (17) of Simson et al. (2021) reveals that this numerical integration implicitly assumes that the strain rate calculated at node $k$ from the stress and viscosity evaluated at node $k$ actually applies to the whole space interval between the node $k$ and the node $k-1$ (assuming that the $\Delta z_j^n = z_{j+1}^n - z_j^n$ stated in Eq. (17) of Simson et al. (2021) actually means $\Delta z_j^n = z_j^n - z_{j-1}^n$, which makes more sense and is in accordance with their published code). Similarly, the numerical integration of the momentum conservation equation yielding the nodal stresses from the distribution of $\Phi_i$ as expressed in Eq. (18) of Simson et al. (2021) implicitly assumes that the value of $\Phi_i$ stored at node $k$ actually applies to the whole space interval between nodes $k$ and $k+1$. In other words, in the approach of Simson et al. (2021), the distribution of $\Phi_i$ is piece-wise constant over numerical layers but the information is only assigned to the bottom node of the considered numerical layer. As detailed in Appendix C1, the fact that $\Phi_i$ at node $k$ relates to the snow mass contained between nodes $k$ and $k+1$, whereas the strain rate calculated at node $k$ is used to deform the space interval between nodes $k$ and $k-1$, leads to an inconsistency that hamper mass conservation.



In contrast, in our approach, $\Phi_i$ is defined element-wise whereas the stress is a nodal variable. The latter is calculated at each node $k$ as the cumulative weight of all overlying elements, which writes:

$$\sigma_{zz,k}^{n+1} = \sum_{j=k}^{N_z-1} g\Phi_{i,j+1/2}^n \rho_i(z_{j+1}^n - z_j^n). \tag{20}$$

This equation is formally equivalent to Eq. (18) of Simson et al. (2021) except that $\Phi_{i,j+1/2}$ explicitly corresponds to an averaged quantity applying to the whole element $j+1/2$ comprised between nodes $j$ and $j+1$. The motion of all nodes can then be determined directly by solving Eq. (19) where the integral term is treated through the Gaussian quadratures.

**Update of ice volume fraction**

It can be shown that conservation of the ice mass is guaranteed only if the temporal discretization of Eq. (16) is based on an implicit numerical scheme (Appendix D). Therefore, in our approach we replace the first-order Euler explicit temporal discretization given in Eq. (15) of Simson et al. (2021) by the following:

$$\Phi_{i,k+1/2}^{n+1} = \frac{\Phi_{i,k+1/2}^n + \Delta t \frac{c_{k+1/2}^{n+1}}{\rho_i}}{1 + \Delta t \dot{\epsilon}_{zz,k+1/2}^{n+1}}, \tag{21}$$

where the mean strain rate $\dot{\epsilon}_{zz,k+1/2}^{n+1}$ and mean deposition rate $c_{k+1/2}^{n+1}$ of element $k+1/2$ are calculated as the integral of the corresponding field over the element using Gaussian quadratures divided by the length of the considered element before deformation.

## 4 Numerical simulations

In this Section, we compare the capabilities of the various numerical treatments introduced above to provide solutions to the coupled problem of heat transport, vapor transport, and settlement in snow that are stable, accurate, and respect energy and mass conservation. To this end, we use some of the experiments proposed by Schürholt et al. (2022) and Simson et al. (2021) as numerical benchmarks. The main features of the numerical set-up and model configurations of all experiments described in the following are summarised in Table 2. All simulations are run with the parameterization (A6) for the saturation vapor density, and with the Calonne parameterizations of effective parameters given by Eqs. (A1)-(A3), unless stated otherwise. Note that the Hansen parameterizations of effective parameters given by Eqs. (A2)-(A4) have also been implemented. In what follows, we use the absolute root-mean-square deviations (RMSDs) as a metric when comparing two solution fields produced with two different implementations. For interpretation, it is important to relate these RMSDs to the typical orders of magnitude of the considered fields shown in respective figures.

### 4.1 On the form of the mass matrices

In this part, we reproduce the scenario 2 presented in Schürholt et al. (2022) to illustrate the importance of dealing with proper mass matrices. Concretely, we consider a $1$ m-thick snowpack with a piece-wise linear initial density profile mimicking a



**Table 2.** Summary of numerical set-ups and model configurations. Scenario 2 and 3 are taken from Schürholt et al. (2022). Case 6 and 7 are taken from Simson et al. (2021). Dirichlet BCs on $T$ are $T_0 = 273$ K and $T_h = 253$ K. Dirichlet BCs on $\rho_v$ are $\rho_{v,0} = \rho_v^{eq}(T_0)$ and $\rho_{v,h} = \rho_v^{eq}(T_h)$.

| Sect. | Set-up | $N_z$ | $\Delta t$ (min) | BC on $T$ | BC on $\rho_v$ | Configuration | Mass matrix[a] | $\alpha$ | Dep.[b] | Settl.[c] |
|---|---|---|---|---|---|---|---|---|---|---|
| 4.1 | Scenario 2 | 201 | 15 | Dirichlet | Dirichlet | FEniCS[d] | I. / N.L. | $5 \times 10^{-3}$ | On | Off |
| | | | | | | CC_3DOF | I./ N. L. | | | |
| | | | | | | CC_3DOF | P./ N. L. | | | |
| | | | | | | CC_3DOF | P./ L. | | | |
| 4.2 | Scenario 2 | 201 | 15 & 5 | Dirichlet | Dirichlet | CC_2DOF | P./ L. | $5 \times 10^{-3}$ | On | Off |
| | | | | Dirichlet | Dirichlet | CC_3DOF | P./ L. | $5 \times 10^{-3}$ | | |
| | | | | Dirichlet | Dirichlet | CD_PC | P./ L. | $5 \times 10^{-3}$ | | |
| | | | | Dirichlet | Dirichlet | CD_FD | P./ L. | $5 \times 10^{-3}$ | | |
| | | | | Dirichlet | No Flux | H_MF | - / L. | - | | |
| | | | | Dirichlet | No Flux | H_TF | P./ L. | - | | |
| 4.3 | Scenario 2 | 201 | 15 | Dirichlet | No Flux | CC_3DOF | P./ L. | $0 \rightarrow 1$ | On | Off |
| | | | | Dirichlet | No Flux | H_MF | -/L. | - | | |
| 4.4 | Scenario 2 | 201 | 15 & 5 | No Flux | No Flux | CC_3DOF | P./ L. | $5 \times 10^{-3}$ | On & Off | Off |
| | | | | | | CD_PC | P./ L. | $5 \times 10^{-3}$ | | |
| | | | | | | H_MF | -/ L. | - | | |
| | | | | | | H_TF | P./ L. | - | | |
| 4.5 | Case 6 | 11 & 51 & 101 | 15 | - | - | Simson code[e] | - | - | Off | On |
| | | | | | | Our code | | | | |
| 4.6 | Case 7 | 101 | Adaptive | Dirichlet | $c = 0$ | Simson code[e] | P./ - | - | On | On |
| | | | 15 | Dirichlet | $c = 0$ | H_MF | -/ L. | - | | |
| | | | 15 | Dirichlet | No Flux | H_MF | -/ L. | - | | |
| 4.7 | Scenario 3 | 1001 | 1 | Dirichlet | Dirichlet | FEniCS[d] | I./ N.L. | $5 \times 10^{-3}$ | On | Off |
| | | | | | | CC_3DOF | I./ N. L. | | | |
| | | | | | | CC_3DOF | P./ L. | | | |
| 4.8 | Scenario 3 | 1001 | 1 | Dirichlet | Dirichlet | CC_3DOF | P. L. | $5 \times 10^{-3}$ | On | On / Off |

[a] I.: Improper, P.: Proper, N.L.: Not Lumped, L.: Lumped (Sect. 3.2).    [b] Deposition.    [c] Settlement.    [d] Schürholt et al. (2022).
[e] Simson et al. (2021).

stratified snowpack containing a crust, as well as an ice layer at the bottom (Eq. (16) of Schürholt et al., 2022). All BCs are of
Dirichlet type and constant over time: the bottom and top temperatures are fixed to, resp., $T_0 = 273$ K and $T_h = 253$ K, while the bottom and top water vapor densities are set assuming saturation at boundaries, i.e. $\rho_{v,0} = \rho_v^{eq}(T_0)$ and $\rho_{v,h} = \rho_v^{eq}(T_h)$.





The initial temperature profile is linear between boundary values, and the initial water vapor density profile is deduced from the latter assuming that water vapor density is initially saturated (solid black lines in Fig. 3). To facilitate a comparison with results presented by Schürholt et al. (2022), all simulations described in this part are run with settlement de-activated and

deposition feedback on $\Phi_i$ activated. The time step is set to $\Delta t = 15$ min. The mesh is uniform and contains 200 elements. For the simulations with the Calonne model, the sticking coefficient is set to a default value of $\alpha = 5 \times 10^{-3}$ so that our $s\alpha v_{\mathrm{kin}}$ is comparable to the $\rho_i s / \beta \rho_v^{eq}$ adopted by Schürholt et al. (2022).

Figure 3 shows the vertical profiles of temperature, vapor density, and deposition rate produced after 38h of simulation by FEniCS on the one hand, and different versions of CC_3DOF on the other hand. The FEniCS run is based on the code

published by Schürholt et al. (2022), with slight adaptations. In particular, the original Crank-Nicholson time scheme adopted by Schürholt et al. (2022) is replaced by a full-implicit scheme. Despite slight differences in the numerical implementation (e.g., $\Phi_i$ is a nodal variable in FEniCS and an element-wise variable in our model, variable-dependent parameters are evaluated at nodes in FEniCS and directly at Gaussian points in our model, the expression of the source terms are not strictly equivalent in FEniCS and in our model), the solution fields produced by FEniCS are very close to those obtained with CC_3DOF when

the mass matrix has the improper form and no lumping is performed (superimposed green solid and red dotted lines). This can be quantified by calculating the RMSDs between the solutions obtained with the two approaches: it is of $3.7 \times 10^{-4}$ K for $T$, and of $4.4 \times 10^{-8}$ kg m$^{-3}$ for $\rho_v$. A more noticeable difference regards the amplitude of the oscillations occurring on the field of $c$ at the top boundary of the domain. Note that oscillations are also observed at the bottom boundary of the domain, as well as at the boundaries of the deposition/sublimation peaks, but they are of similar amplitude in both approaches, except at the

very bottom node. Thus, the RMSD between the fields of $c$ produced by the two approaches is of $3.7 \times 10^{-6}$ kg m$^{-3}$ s$^{-1}$ when the whole domain is considered, and is reduced to $2.4 \times 10^{-8}$ kg m$^{-3}$ s$^{-1}$ when the first bottom node and four top nodes are omitted.

In contrast, adopting the proper form of the mass matrix changes significantly the $T$ and $\rho_v$ fields in places where peaks of sublimation/deposition are observed (blue dashed lines on top and middle panels hidden below orange solid lines): the

RMSDs between the solution fields of $T$ and $\rho_v$ obtained with CC_3DOF using the improper mass matrix and the ones obtained with CC_3DOF using the proper mass matrix are increased to, respectively, $0.42$ K and $6.5 \times 10^{-5}$ kg m$^{-3}$. The sublimation/deposition peaks also become sharper (blue dashed line on bottom panel). The oscillations are mostly reproduced, with larger amplitudes at deposition/sublimation peaks and smaller amplitudes at domain boundaries, especially at the bottom boundary. These spurious oscillations are related to the violation of the DMP. Note that oscillations of the same nature also

occur when FEniCS is run with a Crank-Nicholson time scheme (not shown). As expected, these oscillations vanish when the mass matrix is lumped (orange solid line in Fig. 3).

### 4.2    On the different solver options

Here, we use the same numerical set-up as above in order to compare the relative performances of the various numerical treatments of the Calonne system that are summarised in Fig. 1b. All simulations are run with the proper form of the mass

matrices and with lumping activated. In addition to the time step of $\Delta t = 15$ min, all simulations are also run with a time



**Figure 3.** Comparison of temperature (top), water vapor density (middle) and deposition rate (bottom) fields produced by FEniCS, CC_3DOF without lumping and with the improper mass matrix, CC_3DOF without lumping but with the proper mass matrix, and CC_3DOF with lumping and proper mass matrix after 38h of simulation for the scenario 2 of Schürholt et al. (2022). The blue dashed and orange solid lines are superimposed in the top and middle panels. The solid black lines correspond to the initial conditions.



**Figure 4.** Comparison of temperature (top), water vapor density (middle) and deposition rate (bottom) fields produced by CC_3DOF and CD_PC with $\Delta t = 5$ min and $\Delta t = 15$ min. Solutions are shown after 2 h, 4 h, and 24 h of simulation. Zoom-in over the deposition peaks are included in the bottom panel. Note that green and blue lines are almost superimposed in all panels.

step reduced to $\Delta t = 5$ min. A first important result is that the boundary values taken by the field of $c$ depart strongly from their distribution in the interior of the domain, and that they are highly sensitive to the chosen numerical approach (bottom





panel of Fig. 4). This topic is treated in details in Sect. 4.6. Secondly, it turns out that the two strategies investigated for the coupled solution of the Calonne system, i.e. CC_2DOF and CC_3DOF, produce solution fields that are very close. After $24\,\mathrm{h}$ of simulation with $\Delta t = 15\,\mathrm{min}$, the RMSDs on the $T$, $\rho_\mathrm{v}$, and $c$ fields produced with the two methods amount to, respectively, $1.8 \times 10^{-4}\,\mathrm{K}$, $1.6 \times 10^{-8}\,\mathrm{kg\,m^{-3}}$, and $1.2 \times 10^{-6}\,\mathrm{kg\,m^{-3}\,s^{-1}}$. The higher (relative) deviation on the field of $c$ is mostly due to the values obtained at the two boundary nodes which are slightly sensitive to the chosen strategy: if the very top and bottom nodes are omitted, the RMSD on the $c$-field is reduced to $3.7 \times 10^{-9}\,\mathrm{kg\,m^{-3}\,s^{-1}}$. In addition, the solution fields produced by CC_2DOF and CC_3DOF show very little sensitivity to the time step size (superimposed green and blue lines in Fig. 4, and similar RMSDs as above for all three fields when $\Delta t = 5\,\mathrm{min}$). In contrast, the decoupled solution of the Calonne system leads to significantly different behaviours. First, the CD_FD approach is highly unstable with the $\rho_\mathrm{v}$ and $T$ fields rapidly diverging towards unrealistic values. This continues to be the case even when the time step size is decreased down to $\Delta t = 1\,\mathrm{s}$. In contrast, the CD_PC approach gives steady solutions that are mostly independent of the time step. These steady solutions are not very different from those obtained with the coupled approaches (solid red and orange lines in Fig. 4). Concretely, after $24\,\mathrm{h}$, the RMSDs between the solution fields produced with CC_3DOF and CD_PC are of $4.2 \times 10^{-2}\,\mathrm{K}$, $4.8 \times 10^{-6}\,\mathrm{kg\,m^{-3}}$, and $4.3 \times 10^{-5}\,\mathrm{kg\,m^{-3}\,s^{-1}}$ ($2.4 \times 10^{-6}\,\mathrm{kg\,m^{-3}\,s^{-1}}$ when top and bottom nodes are omitted) for the fields of $T$, $\rho_\mathrm{v}$, and $c$, respectively. Yet, the transient solution fields produced by CD_PC show oscillations (even when the mass matrices are lumped) and high sensitivity to the time step size (dotted and dashed red and orange lines in Fig. 4). For example, after $2\,\mathrm{h}$ of simulation, RMSDs between CD_PC solutions obtained with $\Delta t = 15\,\mathrm{min}$ on the one hand, and with $\Delta t = 5\,\mathrm{min}$ on the other hand, amount to $1.3\,\mathrm{K}$, $1.7 \times 10^{-4}\,\mathrm{kg\,m^{-3}}$, and $7.2 \times 10^{-4}\,\mathrm{kg\,m^{-3}\,s^{-1}}$ ($2.4 \times 10^{-5}\,\mathrm{kg\,m^{-3}\,s^{-1}}$ when top and bottom nodes are omitted) for, respectively, the fields of $T$, $\rho_\mathrm{v}$, and $c$. This behaviour has to be compared to the transient solutions produced by CC_3DOF/CC_2DOF (blue lines in Fig. 4) that are smooth and smoothly converge to the steady solutions.

The much higher stability of CD_PC compared to CD_FD is due to the proper treatment of the reaction term in the water vapor mass balance equation, which acts as an attractor of $\rho_\mathrm{v}$ toward $\rho_v^{eq}$. Nevertheless, the results presented in this paragraph underline the major importance of having a coupled solution of the coupled heat and water vapor diffusion equations. Although the CD_PC approach gives steady solutions that are close to the ones obtained with the coupled approaches, accuracy of the transient responses is essential as the external forcings are constantly evolving in time for the vast majority of real-world configurations.

The same experiment has been run with the configurations H_MF and H_TF (not shown). The two strategies turn out to produce solution fields of $T$ (and consequently of $\rho_\mathrm{v}$ and $c$ as both fields are diagnosed from the obtained $T$-field) that are very close to each other, and that show very little sensitivity to the time step size: the RMSD between both strategies after $24\,\mathrm{h}$ of simulation with $\Delta t = 5\,\mathrm{min}$ (resp. $\Delta t = 15\,\mathrm{min}$) is of $6.2 \times 10^{-3}\,\mathrm{K}$ (resp. $6.2 \times 10^{-3}\,\mathrm{K}$), $5.6 \times 10^{-7}\,\mathrm{kg\,m^{-3}}$ (resp. $5.5 \times 10^{-7}\,\mathrm{kg\,m^{-3}}$), and $7.3 \times 10^{-9}\,\mathrm{kg\,m^{-3}\,s^{-1}}$ (resp. $7.3 \times 10^{-9}\,\mathrm{kg\,m^{-3}\,s^{-1}}$) for the fields of $T$, $\rho_\mathrm{v}$, and $c$, respectively. As we shall see in Sect. 4.4, only the H_MF approach is perfectly energy conservative.



**Figure 5.** Comparison of temperature (top), water vapor density (middle) and deposition rate (bottom) fields produced after 38h of simulation with H_MF on the one hand, and CC_3DOF for various values of $\alpha$ on the other hand. Note that the case $\alpha = 1$ is not represented as it would not be distinguishable from the case $\alpha = 10^{-1}$.



**Table 3.** RMSDs between solutions obtained with H_MF on the one hand, and with CC_3DOF for the various values of $\alpha$ on the other hand

| RMSD on: | $T$ (K) | $\rho_\mathrm{v}$ (kg m$^{-3}$) | $c$ (kg m$^{-3}$ s$^{-1}$) |
|---|---|---|---|
| $\alpha = 0$ | $4.3 \times 10^{-2}$ | $1.1 \times 10^{-3}$ | $1.5 \times 10^{-6}$ |
| $\alpha = 10^{-8}$ | $2.1 \times 10^{-2}$ | $6.8 \times 10^{-5}$ | $1.4 \times 10^{-6}$ |
| $\alpha = 10^{-6}$ | $8.1 \times 10^{-3}$ | $1.6 \times 10^{-6}$ | $9.3 \times 10^{-7}$ |
| $\alpha = 10^{-4}$ | $1.1 \times 10^{-2}$ | $1.0 \times 10^{-6}$ | $6.7 \times 10^{-8}$ |
| $\alpha = 10^{-1}$ | $1.1 \times 10^{-2}$ | $1.0 \times 10^{-6}$ | $9.4 \times 10^{-9}$ |
| $\alpha = 1$ | $1.1 \times 10^{-2}$ | $1.0 \times 10^{-6}$ | $9.4 \times 10^{-9}$ |

## 4.3 Comparing the Hansen and Calonne systems for high $\alpha$

Figure 5 shows the solutions produced after 38h with our model in the configurations H_MF on the one hand, and CC_3DOF for various values of the sticking coefficient $\alpha$ on the other hand. All simulations are run with the proper form of the mass matrices from the numerical set-up introduced above, except for the BCs on $\rho_\mathrm{v}$ which are forced to no flux in all cases (Table 2) so that both configurations are easily comparable (see Sect. 4.6). For both configurations, effective parameters are computed from the Calonne parameterizations (Appendix A). We recall that the Calonne system has been derived through the two-scale expansion method which is valid for low reaction rates only, i.e. $\alpha \sim 10^{-3}$ or less (Sect. 2.1). Therefore, neither the Calonne system nor the Calonne parameterization of effective parameters are expected to be valid for higher values of $\alpha$. Yet, our goal here is simply to compare the mathematical behaviours of the Hansen and Calonne systems when they are run with the same parameterizations as done by Schürholt et al. (2022), independently of the physical soundness of obtained results.

The solution fields produced by CC_3DOF progressively converge to the ones produced by H_MF as $\alpha$ increases and a higher reaction rate forces the vapor to saturation. This behaviour was expected. Indeed, if effective parameters are calculated the same way and BCs are identical, then both systems are formally equivalent when water vapor density is assumed to always be at saturation (Sect. 2.3). To quantify this behaviour, Table 3 gathers all RMSDs between solutions produced at the end of the simulation by H_MF on the one hand, and CC_DOF for all tested values of $\alpha$ on the other hand. The difference in $\rho_\mathrm{v}$ and $c$ between H_MF and CC_3DOF becomes less than 1% when $\alpha$ becomes higher than $10^{-6}$ and $10^{-3}$, respectively. Below these values of $\alpha$, the difference varies from a few percent to $\sim 100\%$ as $\alpha$ tends to 0. In contrast, $T$ is less affected by the value of $\alpha$. Even for the lowest values tested, the differences between the fields of $T$ produced by H_MF and CC_3DOF are below 0.1% (all lines are superimposed on top panel of Fig. 5).

All together, these results seem to contradict the ones presented in Fig. 2 of Schürholt et al. (2022): in their case, the solution fields (especially $c$) produced by the Hansen and Calonne models differ significantly, even when both models rely on the Calonne parameterizations for effective parameters. We think that this is due to the improper treatment of the mass matrices in their FEniCS implementation. This conclusion is supported by the fact that "existing continuum-mechanical models derived through homogenisation (i.e., Calonne) or mixture theory (i.e. Hansen) yield similar results for homogeneous snowpacks of constant density" (Schürholt et al., 2022), as demonstrated from their scenario 1. Indeed, when density is uniform, the



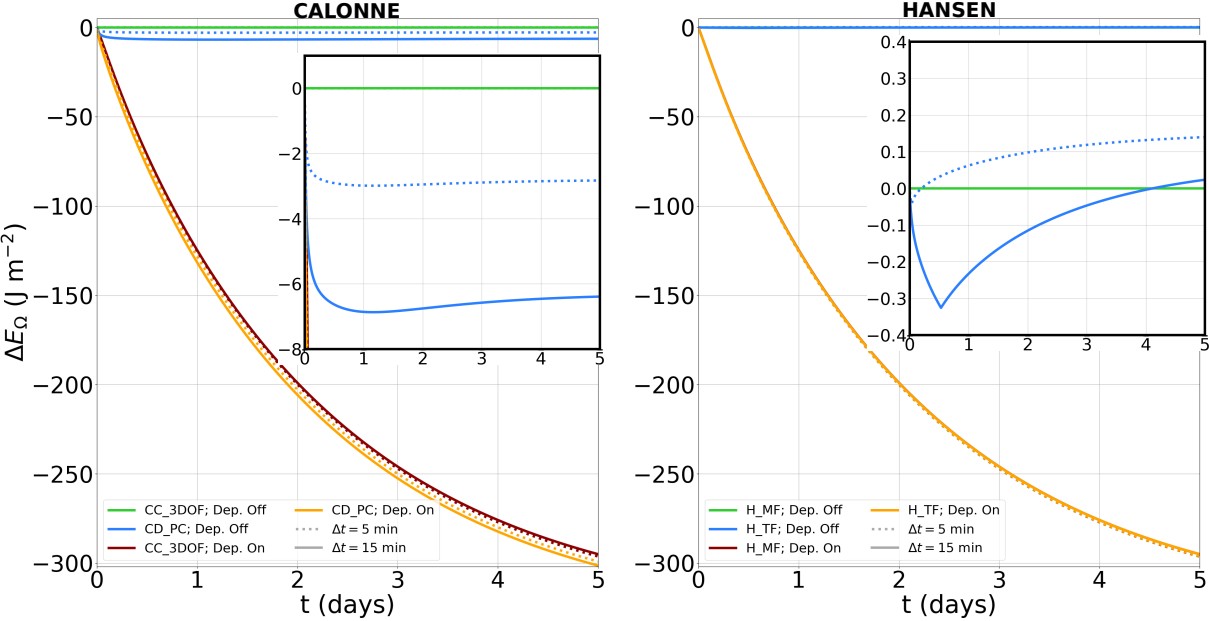

**Figure 6.** Energy leakage as a function of simulation time for the Calonne (left panel) and Hansen (right panel) systems, with and without deposition. For the Calonne system, we test the strategies CC_3DOF and CD_PC. For the Hansen system, we test the strategies H_MF and H_TF. Each simulation is run with $\Delta t = 5$ min and $\Delta t = 15$ min. Note that on right panel, orange and red lines are almost superimposed. Both panels contain a zoom-in for the region around $\Delta E_\Omega = 0$ J m$^{-2}$.

multiplying factors of the accumulation terms are independent of space and can be arbitrarily affected directly to the mass matrix or their inverses to the stiffness matrix and force vector.

## 4.4   On energy conservation

In order to evaluate how the different numerical strategies considered behaves for what regards energy conservation, we run a slightly modified version of the numerical set-up introduced in Sect. 4.1: the original Dirichlet BCs are replaced by no flux
BCs for both heat and vapor. Because settlement is de-activated (no vapor expelled from the snowpack to the atmosphere), the system is thus closed and all recorded energy leakage should be considered as numerical artefacts. We run the experiment for 5 days, testing all strategies of the three options presented in Sect. 3.2 and summarised in Fig. 1b, except the CD_FD which yields unrealistic results even for time step as low as $\Delta t = 1$ s. All experiments are run twice, once with $\Delta t = 15$ min and another time with $\Delta t = 5$ min. Then, the feedback of deposition on $\Phi_i$ is de-activated and the procedure is repeated. The obtained energy
leakage are represented as a function of time in Fig. 6. Again, results produced with CC_3DOF are very close to those obtained with CC_2DOF, and only the former are reported. The total energy leakage at the end of the simulation are summarised in Table 4. As expected, decoupling the deposition-related evolution of $\Phi_i$ (source term of the ice mass conservation equation) from the heat/vapor transfer equation induces an artificial energy loss which amounts to $\sim 300$ J m$^{-2}$ for all considered cases





**Table 4.** Total energy leakage (J m$^{-2}$) after 5 days of simulation.

| Deposition: | off | | on | |
|---|---|---|---|---|
| $\Delta t =$ | 5 min | 15 min | 5 min | 15 min |
| CC_3DOF | 0 | 0 | -296.3 | -295.0 |
| CC_PC | -2.8 | -6.4 | -299.1 | -301.4 |
| H_MF | 0 | 0 | -296.3 | -295.0 |
| H_TF | 0.14 | 0.02 | -296.2 | -295.0 |

at the end of the simulation. This energy loss remains very limited compared to the total energy contained in the system but

could become noticeable for simulations on seasonal timescales and/or in configurations associated to stronger deposition rates. On the contrary, when deposition is omitted, energy leakage become negligible. Nevertheless, we stress that, contrary to the strategies CC_2DOF, CC_3DOF and H_MF that are rigorously conservative, the strategies CD_PC and H_TF induce tiny but non-zero energy leakage, in line with the demonstration of Tubini et al. (2021). This illustrates that a rigorously energy-conservative solution of the problem of heat conduction and water vapor diffusion in snow implies to solve the heat diffusion,

water vapor diffusion and ice mass conservation equations in a coupled way for the Calonne system, and to opt for the mixed form of the heat equation for the Hansen system.

### 4.5 On mass conservation

In this part, we reproduce the numerical set-up corresponding to the case 6 designed by Simson et al. (2021): we consider a snowpack with an initial thickness of $50 \, \text{cm}$ split into two equally thick snow layers of uniform initial densities, i.e. $150 \, \text{kg m}^{-3}$

for the lower one and $75 \, \text{kg m}^{-3}$ for the top one. All simulations are run considering only the settlement process. Our goal is to illustrate how the modifications made to the settlement scheme of Simson et al. (2021) affect the mass conservation and settlement rates. For this, we run the code published by Simson et al. (2021) with three bug fixes that are described in Apppendix C. In particular, the strain rate governing the deformation of the whole space interval comprised between nodes $k$ and $k+1$ is calculated from the stress and viscosity evaluated at node $k$ (and not at node $k+1$ as in the published version

of the code, Appendix C1). Simulations are run for 20 days, with $\Delta t = 15 \, \text{min}$, and with three initially uniform meshes containing, respectively, 11, 51, and 101 nodes. For all runs, viscosity is calculated from Eq. (A7) with a fixed temperature set to $T = 263 \, \text{K}$.

We first run the simulations using the original explicit time discretization scheme implemented by Simson et al. (2021) to update $\Phi_i$. As illustrated in Fig. 7a, mass is not conserved in this case: there is an artificial mass loss that gets higher for

coarser meshes. In the present case, this mass loss is of the order of 20 g after 20 days of simulation for a total initial mass of 56.25 kg. Although this mass loss tends to stabilise after a few days as viscosity increases exponentially with increasing $\Phi_i$, it will become more significant when integrated over a full season characterised by regular snowfalls of low initial densities. As soon as the explicit time integration scheme is replaced by an implicit one, i.e. following Eq. (21) and Appendix C4, mass

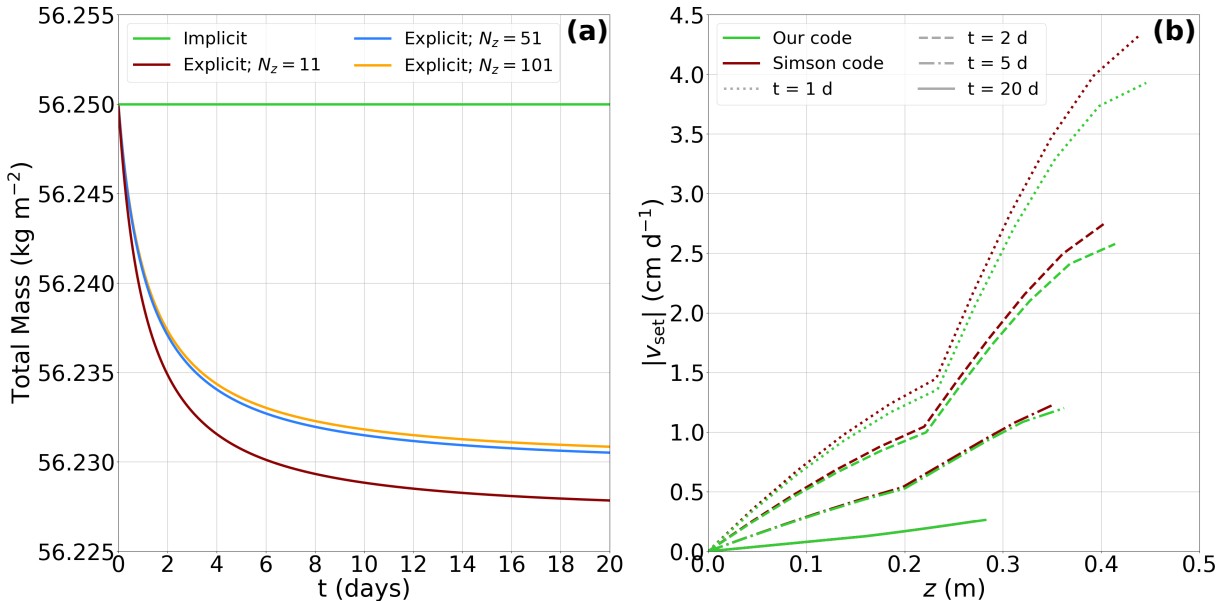

**Figure 7.** Panel (a) shows the evolution of the total mass over time when $\Phi_i$ is updated using the original explicit time discretization scheme implemented by Simson et al. (2021) for meshes containing 11, 51, and 101 nodes. As soon as the time discretization scheme becomes implicit, mass is conserved independently of mesh refinement (green line). Panel (b) shows the absolute settling velocity profile obtained on the 11-nodes mesh with our model, and with the corrected code of Simson et al. (2021) using an implicit time discretization scheme after 1, 2, 5, and 20 days of simulation.

is perfectly conserved independently of the mesh size. Note that this is the case when running both a corrected version of the
code published by Simson et al. (2021) in which the explicit settlement scheme has been replaced by an implicit one, and our
model (green line in Fig. 7a). On Fig. 7b, we compare the settlement rates obtained with the code of Simson et al. (2021)
using the implicit time integration scheme to the one obtained with our code for a 11-nodes mesh. In the approach of Simson
et al. (2021) corrected following Appendix C1 to ensure conservation of mass, the deformation of the whole numerical layer
between $k$ and $k+1$ is implicitly calculated from the stress evaluated at the bottom node of the layer, where it is maximal.
In contrast, we assess the deformation occurring between nodes $k$ and $k+1$ by integrating the ratio between the stress and
viscosity fields along the element $k+1/2$. As a consequence, the former treatment tend to slightly overestimate the settlement
rate compared to ours. This sensitivity is stronger at the beginning of the simulation when settlement rates are higher, and tend
to decrease over time. As expected, when the mesh resolution is increased, the two approaches converge to the same settlement
rates (not shown).

**4.6    On the boundary conditions on water vapor density**

In this part, we want to highlight the sensitivity of the solution fields to the treatment of water vapor at boundaries. To this
end, we reproduce the case 7 proposed by Simson et al. (2021). The latter corresponds to the same numerical set-up as for



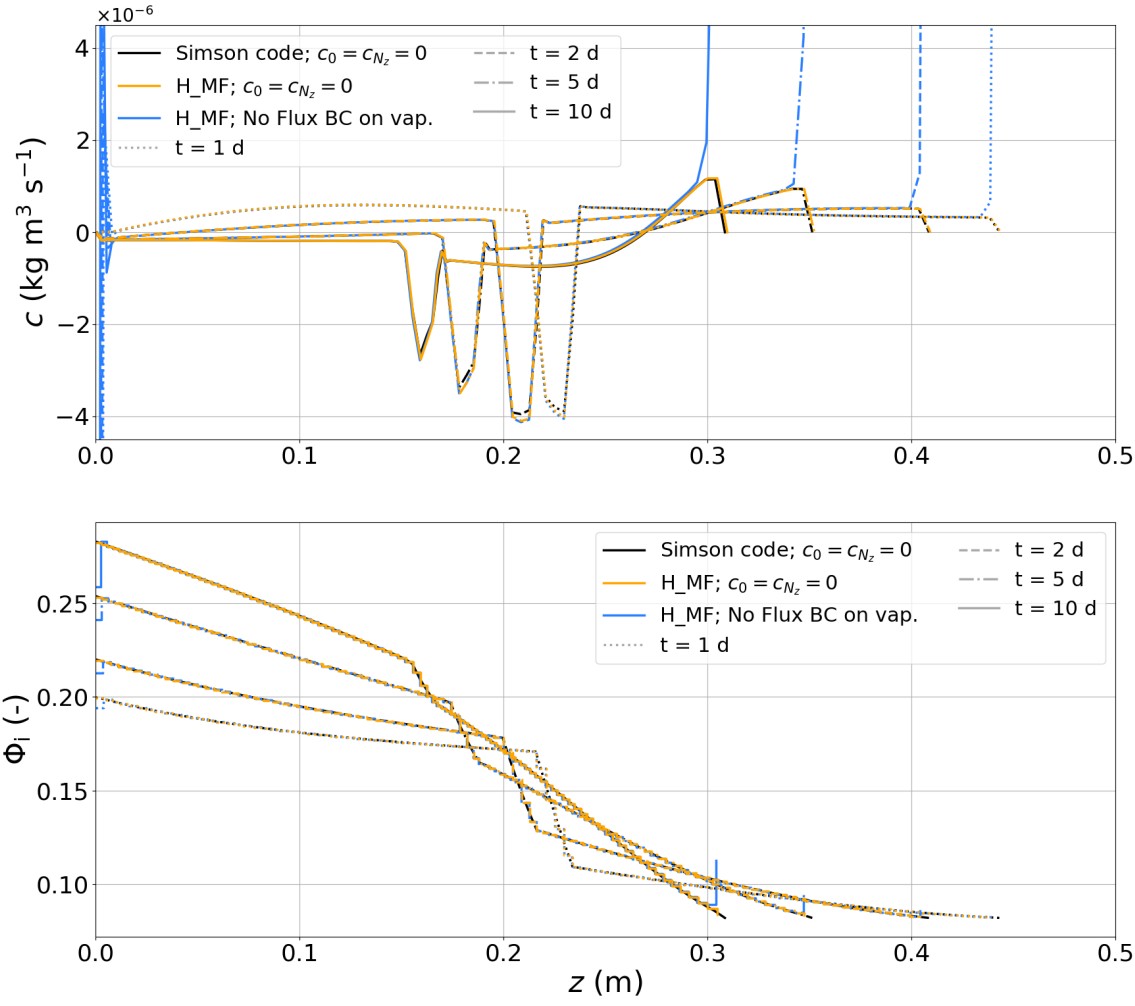

**Figure 8.** Comparison of deposition rate (top) and ice volume fraction (bottom) fields produced after 1, 2, 5, and 10 days of simulation on a 101-nodes mesh running: (i) the corrected version of the code published by Simson et al. (2021); (ii) H_MF with boundary values of $c$ forced to zero; and (iii) H_MF with no constraint on boundary values of $c$ but with a no flux BC on vapor. Note that $\Phi_i$ being an element-wise variable in our approach, it is represented as steps for the two H_MF cases.

the case 6 introduced above, except that the heat and water vapor diffusion processes are included. Furthermore, the feedback of temperature on viscosity is taken into account through Eq. (A7). Figure 8 shows the $c$ and $\Phi_i$ fields obtained after 1, 2, 5

and 10 days of simulation on a 101-nodes mesh when running the code published by Simson et al. (2021) with an implicit settlement scheme and with the bug fixes described in Appendix C (black lines, top panel). The values of $c$ are always zero at the boundaries because they are forced as such in the model by the authors. Despite slight differences in the numerical implementation (e.g., element-wise vs nodal nature of $\Phi_i$, treatment of strain rate, mixed-form vs T-form for the Hansen system), running our model in its H_MF configuration gives solution fields that are consistent with the ones produced by the





code of Simson et al. (2021) if we also force the boundary values of $c$ to zero (orange lines, top panel). Concretely, the RMSDs on $c$ varies between $2.8 \times 10^{-8}$ kg m$^{-3}$ s$^{-1}$ and $3.4 \times 10^{-8}$ kg m$^{-3}$ s$^{-1}$, which corresponds to differences of the order of $\sim 1\%$.

As stated in Sect. 3.2, the deposition rate being derived as the closure of the water vapor mass balance, imposing a vanishing deposition rate at a boundary is equivalent to impose a boundary vapor flux that adjusts itself so that local disequilibrium between $\rho_v$ and $\rho_v^{eq}$ are entirely compensated without any requirements for phase change. This choice is arbitrary and cannot be used as the generic BC for snowpack models. Ideally, the BCs for the vapor mass budget should be derived through a vapor budget at interfaces (akin to the surface energy budget routinely used as BCs for the energy equation). The importance of the BCs on vapor can be illustrated by running the same H_MF simulation as above, but removing all constraint on $c$ and assuming no vapor fluxes at boundaries. The resulting field of $c$ is then characterised by peak values at the two boundary nodes, that depart significantly from the distribution of $c$ in the interior of the domain (blue lines, top panel). These high deposition rates at domain boundaries have direct implications on the field of $\Phi_i$: a significant part of the ice mass of the bottom element sublimates and is transported upwards. This leads to a situation in which the bottom element is less dense than the one immediately above, a tendency that is exacerbated over time (blue lines, bottom panel). This behaviour creates strong local density gradients that seem to trigger self-amplifying oscillations at the bottom boundary: a sublimation peak at the bottom node is followed by a deposition peak at the node right above. This oscillatory pattern is propagating inwards over a number of nodes that is growing in time. On the top, strong mass gain occurs associated to a high deposition peak over the top element, but is not triggering oscillations at this point in time. We stress that the peaks of $c$ at boundaries and oscillations at the bottom boundary are not numerical artefacts but are truly part of the solution of Eq. (3) without vapor fluxes at the boundaries. They must be seen as the deposition rates that are required so that the Hansen system assumption regarding the permanent and instantaneous restoration of the water vapor density to its saturated value is fulfilled when no contribution can be expected from water vapor mass fluxes at the domain boundaries to bridge the gap.

### 4.7 On the formation of density wave instabilities

In their work, Schürholt et al. (2022) have illustrated on a dedicated numerical set-up, and confirmed through linear stability analysis, that both the Hansen and Calonne systems are prone to produce self-amplifying spatial oscillations on the $c$ and $\Phi_i$ fields when regions of very high density gradients are present. These oscillations are true mathematical features related to the dependence of the effective heat and vapor diffusion coefficients on the ice volume fraction (Schürholt et al., 2022). Here, we reproduce their experiment to investigate how this instability materialises when $\Phi_i$ is treated as an element-wise variable rather than as a nodal variable. The numerical set-up consists on a 2-cm thick snowpack with an initial ice volume fraction mimicking a Gaussian crust following Eq. (20) of Schürholt et al. (2022) (solid black line in bottom left panel of Fig. 9). The ICs and BCs on $T$ and $\rho_v$ are the same as those used in Sect. 4.1. For these simulations, the time step is decreased to $\Delta t = 1$ min. Figure 9 shows the $c$ and $\Phi_i$ fields obtained after 48h of simulation on a 1000-nodes mesh. Let's compare first the solutions produced by FEniCS with an implicit time scheme (green solid line) and CC_3DOF with the same (improper) form of the mass matrix as for the FEniCS run and without lumping (dark red dotted line). Consistently with results already reported in Sect. 4.1, solutions



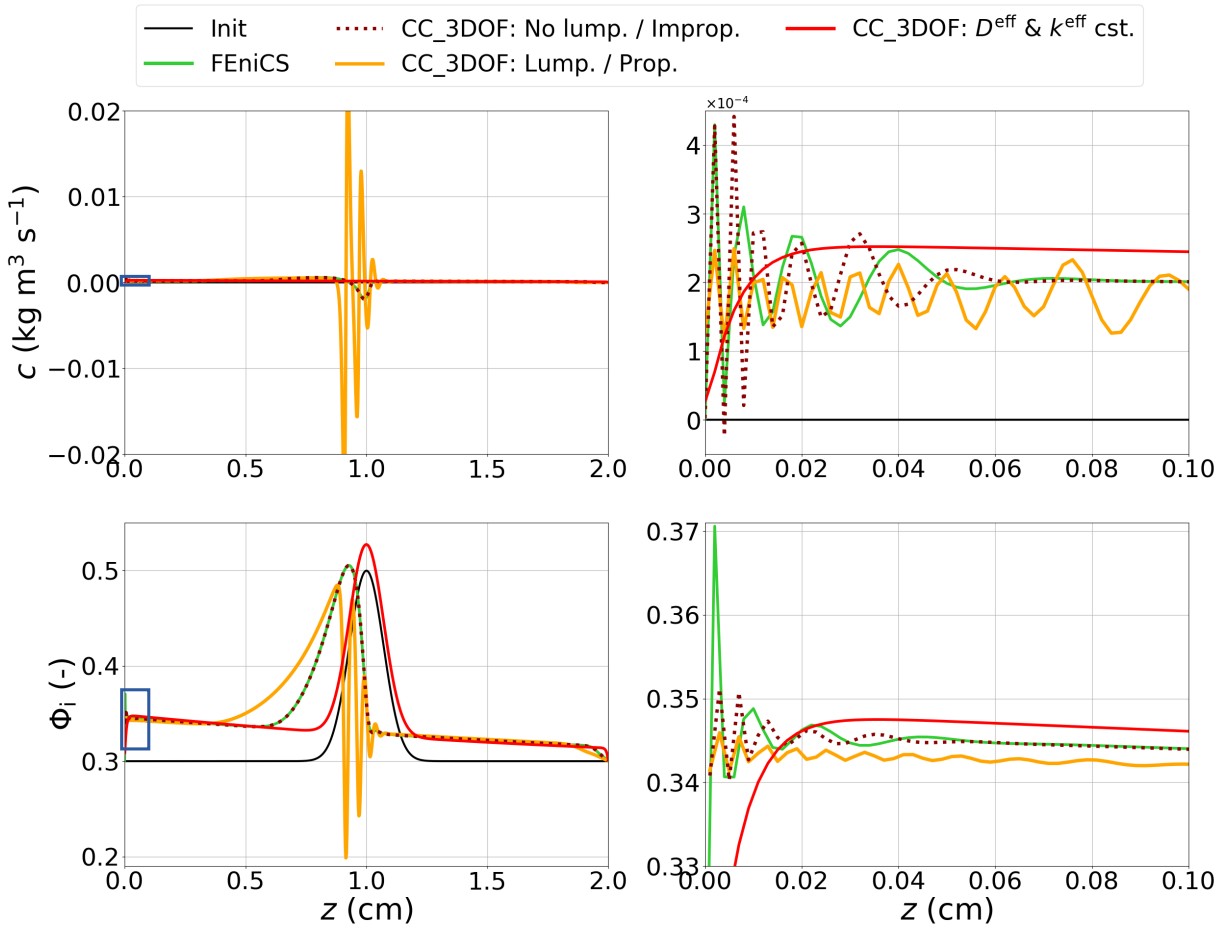

**Figure 9.** Comparison of deposition rate (top panels) and ice volume fraction (bottom panels) fields produced after 2 days of simulation on a 1000-nodes mesh by FEniCS, CC_3DOF without lumping and with the improper mass matrix, and CC_3DOF with lumping and proper mass matrix. The latter is run a second time with constant $D^{\mathrm{eff}}$ and $k^{\mathrm{eff}}$. Blue boxes on left panels highlight zoomed-in areas depicted on right panels. The solid black lines correspond to the initial conditions. Note that despite the element-wise nature of $\Phi_{\mathrm{i}}$ in our model, the latter is represented as lines drawn from elemental values affected to the middle of elements in order to ease comparison with the FEniCS solution.

obtained in both simulations are very close to each other. Concretely, RMSDs on $c$ and $\Phi_{\mathrm{i}}$ are of $1.7 \times 10^{-5}$ kg m$^{-3}$ s$^{-1}$ and

$4.5 \times 10^{-4}$, respectively. In addition, both approaches produce smooth wave patterns - both on $c$ and $\Phi_{\mathrm{i}}$ - encompassing several nodes at the bottom boundary. A closer look in this area (right panels of Fig. 9) reveals slight differences. In particular, the amplitude of the oscillations on $\Phi_{\mathrm{i}}$ are larger on the FEniCS solution than on the CC_3DOF one. This is not surprising given the nodal nature of $\Phi_{\mathrm{i}}$ in the FEniCS approach: a deposition/sublimation peak at a given node translates directly into a ice volume fraction increase/decrease at the same node. In contrast, in our approach the averaging of the nodal $c$ over the elements when updating $\Phi_{\mathrm{i}}$ (sect. 3.3) tends to limit the amplitude of these oscillations, but does not erase them. We also note that,

contrary to results reported in the previous section, no strong sublimation/deposition peaks are observed at the very bottom/top





nodes. In fact, the deposition rate is very slightly positive at both nodes. This is because the Dirichlet BCs $\rho_{\mathrm{v}} = \rho_v^{eq}$ applied on the water vapor mass balance equation of the Calonne system imply water vapor fluxes at boundaries that contribute to maintain the saturation at these nodes, limiting the necessity for deposition/sublimation to bridge the over/under-saturation.

Running the same CC_3DOF simulation with the proper form of the mass matrix (orange solid line in Fig. 9) changes considerably the profiles of $\Phi_{\mathrm{i}}$ and $c$ obtained after 2 days of simulation. In particular, large oscillations on $\Phi_{\mathrm{i}}$ related to large peaks of sublimation and deposition are observed on the sublimating (cold) side of the Gaussian crust. In fact, the general pattern of formation and propagation of the instability wave is not changed compared to the two previous simulations (as illustrated in Fig. 5 of Schürholt et al. (2022)), but oscillations set up at a much faster pace in this simulation. Again, this shows

that treating mass matrices in their improper form can dramatically affect the obtained solutions at a given point in time.

    We run an additional simulation, which consists again in the CC_3DOF strategy with the proper form of the mass matrix and lumping activated, but using constant values for the effective parameters $k^{\mathrm{eff}}$ and $D^{\mathrm{eff}}$. More specifically, we set $k^{\mathrm{eff}} = 0.2 \ \mathrm{Wm}^{-1}\mathrm{K}^{-1}$ and $D^{\mathrm{eff}} = 1.066 \times 10^{-5} \ \mathrm{m}^2\mathrm{s}^{-1}$. This corresponds to the Calonne parameterization of these effective parameters for a $\Phi_{\mathrm{i}}$ fixed to its initial averaged value, i.e. $\Phi_{\mathrm{i}} = 0.318$. The obtained solution shows a non-uniform deposition

over the whole domain (solid red line, top panels). This is because the water vapor fluxes at boundaries implied by the previously mentioned Dirichlet BCs on water vapor bring enough water vapor mass from the exterior so that the whole layer is over-saturated for the implemented values of $k^{\mathrm{eff}}$ and $D^{\mathrm{eff}}$, which causes deposition. In contrast, when the same simulation is run with a no flux condition on vapor at boundaries, we observe a very strong peak of sublimation at the bottom node and a less pronounced peak of deposition at the top node (not shown). The non-uniform deposition leads to an increase of $\Phi_{\mathrm{i}}$ over

the whole domain, more pronounced on the lower half than on the upper half. Logically, the quasi-advection of the Gaussian crust towards the warm boundary that is observed in all other simulations and thoroughly analysed by Schürholt et al. (2022) does not occur here. Indeed, the later is due to continuous deposition on the lower side of the crust associated to continuous sublimation on the upper side, which does not happen in this case. Another remarkable feature is the absence of the wave instability pattern in this case, in line with prediction of Schürholt et al. (2022).

All together, these observations confirm the assertion of Schürholt et al. (2022): the wave patterns on $c$ and $\Phi_{\mathrm{i}}$ are intrinsic features of the mathematical models rather than due to, e.g., the form of the implemented mass matrix or a violation of the DMP; they are triggered by strong density gradients and are due to the dependence of $k^{\mathrm{eff}}$ and $D^{\mathrm{eff}}$ to ice volume fraction. There are slightly smoothen, but not removed, when the $\Phi_{\mathrm{i}}$ is treated as an element-wise quantity rather than as a nodal quantity.

    Note that we have also run the same experiment with the H_MF model using the Hansen parameterization of effective

parameters (not shown). Again our results are in line with those of Schürholt et al. (2022): the wave instability appears much faster and with much stronger amplitude than for the Calonne model with the Calonne parameterization of effective parameters.

### 4.8   On the effect of settlement on the density wave instabilities

We take advantage of our developments to run the same experiment as above, but with settlement activated. The considered domain being only 2 cm-thick and the initial mean density being of 291 $\mathrm{kg \ m}^{-3}$, the stresses are too low and the initial

viscosity too high (viscosity increases exponentially with density) to induce significant settlement in affordable computational



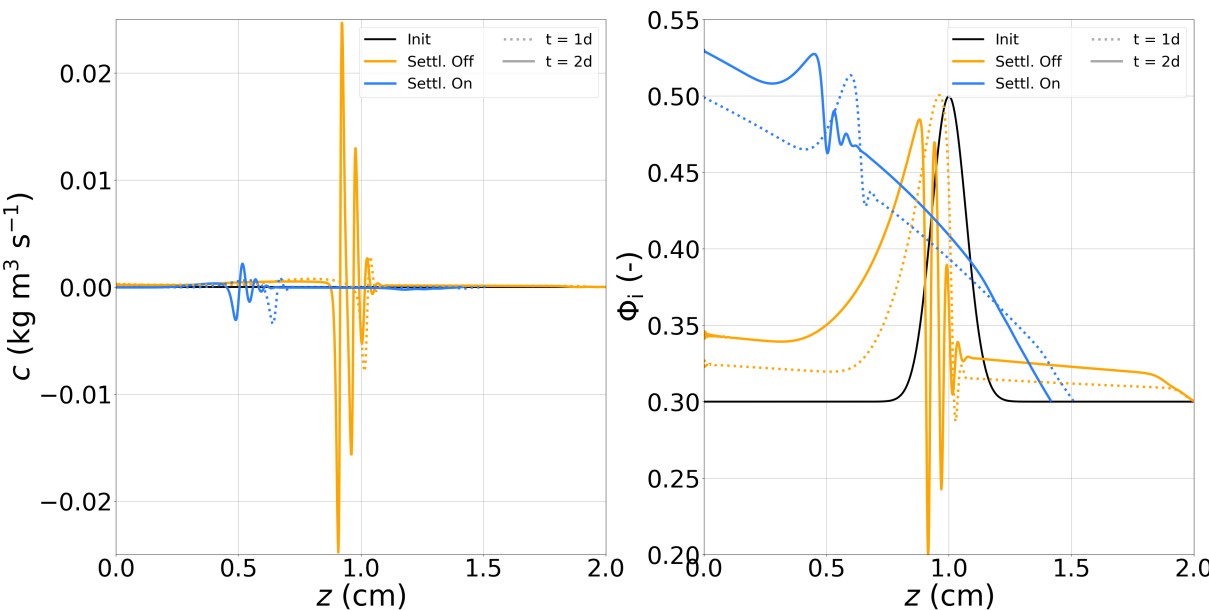

**Figure 10.** Comparison of deposition rate (left) and ice volume fraction (right) fields produced after 1 and 2 days of simulation on a 1000-nodes mesh by CC_3DOF with lumping and proper mass matrix in two configurations: without and with settlement. The solid black lines correspond to the initial conditions. We opt for the same representation of $\Phi_i$ as for Fig. 9.

time if the parameterization (A7) of viscosity is taken as such. Therefore, we decide to keep this parameterization, but to divide the obtained viscosity by a factor $10^4$. Given the linear nature of the implemented constitutive law, the general pattern of deformation is expected to be kept, but with a strongly enhanced settlement rate. We limit the simulations to two runs, both performed with the CC_3DOF strategy with the proper form of the mass matrix, and lumping activated: one is run with

settlement on, the other with settlement off. As illustrated in Fig. 10, including settlement has a stabilising effect on the wave instability: oscillations on $c$ and $\Phi_i$ are still observed but with strongly reduced amplitudes. For the considered numerical set-up, the ratio between the maximum amplitudes of the oscillations observed on $c$ (resp. on $\Phi_i$) when settlement is included and when it is omitted, is of $\sim 40\%$ (resp. $\sim 30\%$) after one day, and of $\sim 20\%$ (resp. $\sim 25\%$) after two days. These oscillations also take longer to set up when considering settlement: after one day of simulation, they are only starting to affect the $c$-field

at the bottom nodes, and their feedback on the $\Phi_i$-field in this area is hardly perceptible. In contrast, when settlement is not included, the oscillatory pattern at the bottom boundary is already well-established on both fields after one day of simulation. This stabilising effect could be expected from the $\Phi_i$-dependence of viscosity: the denser the element, the less it deforms, and reciprocally. This tends to homogenise the density of the snowpack, and competes with phase change which, as shown previously, tends to further densify the denser layers, and to further deplete the hollower layers. Another interesting feature is

that, while only deposition is observed far enough from the Gaussian crust when settlement is off, these areas show sublimation when settlement is included.



# 5    Implications for future developments

The numerical issues that have been highlighted in the present work are relevant beyond the particular processes investigated here. First of all, it is a very general rule that the discretization of a continuous PDE must preserve its properties. This supposes

first to include any factor affecting the accumulation term in the corresponding mass matrix, and not its inverse in the stiffness matrix, as soon as such factor is non-uniform. Second, this contraindicates the use of the chain rule on the accumulation term during time discretization, as it might break the conservative properties of the continuous PDE. In fact, in many situations it is preferable to treat such a PDE in its mixed form, i.e. with the time-derivative of the accumulation term directly applying to the conserved quantity. This result is a generic results for conservation equations, and has already been illustrated on several other

processes (e.g., Celia et al., 1990; Casulli and Zanolli, 2010; Tubini et al., 2021). We have also shown that two-way coupled PDEs need to be solved as single monolithic process to preserve their conservative properties. In addition, depending on the relative dynamics of the sub-processes at stake, solving the latter sequentially might require to considerably decrease the time step in order to guarantee acceptable transient solutions. A monolithic treatment of coupled equations does not fundamentally increases the complexity and numerical cost of the solution. Indeed, solving two decoupled tri-diagonal systems of equations

or solving one coupled penta-diagonal system of equations both come with the same complexity (i.e. $\mathcal{O}(n)$; El-Mikkawy and Atlan, 2014; Jia and Jiang, 2015). The possibility to use larger time steps with a monolithic treatment is thus beneficial in terms of computational cost. Again, this result hold as soon as two-way coupled PDEs are at stake, in snow modelling or elsewhere.

More generally, coming up to a numerical implementation that guarantees energy and mass conservation is not straightforward. Therefore, a rigorous assessment of the evolution of energy and mass within the domain is a golden rule that should guide

any numerical developments. This requires to know the energy and mass fluxes in and out of the domain, and thus to explicitly compute the residuals of the algebraic system. In the eventuality that constraint on numerical cost requires numerical implementations violating the energy/mass conservation, then the associated leakage should remain negligible, be well-identified, and be tracked carefully. This is what we have done for the energy leakage associated to the sequential treatment of the feedback of deposition/sublimation on the ice mass conservation. Not only is this required to ensure that the model in itself produces

physically-sound results, but also in the perspective of latter couplings to other components of Earth System Models (ESMs). Indeed, the introduction within the latter of artificial sink/source of energy/mass obviously contributes to undesired model drift (e.g., Gupta et al., 2013; Hobbs et al., 2016). This drift is all the easier to correct as the spurious energy/mass leakage are well identified and correctly quantified.

More specific to the problem of water vapor diffusion is the treatment of water vapor at boundaries. We have shown in Sect.

4.6 how sensitive the solution fields can be to this treatment. Previous works modelling macroscopic water vapor diffusion in snow have only considered two types of BCs on vapor: a no flux condition (e.g., Touzeau et al., 2018), or a Dirichlet-type BC enforcing saturation of vapor at boundaries (e.g., Schürholt et al., 2022). Jafari et al. (2020) used a mix of both, i.e. no flux at the bottom BC and saturation at the top BC. We also recall that forcing the deposition rate to zero at boundary nodes when solving the Hansen system, as done by Simson et al. (2021), is a roundabout way to arbitrarily prescribe vapor fluxes at boundaries

that perfectly compensate the over/under-saturation at these nodes. In natural configurations, vapor fluxes are expected at both





boundaries. It follows that the proper way to proceed for real-world application is to solve a surface water vapor mass balance in order to derive well-defined boundary water vapor fluxes, similarly to what is normally done for energy. In contrast, cold laboratory experiments can impose a no flux condition on vapor at boundaries by using impermeable plates. Such experiments reveal strong sublimation at the bottom BC, and strong deposition at the top BC (e.g., supplement of Hagenmuller et al., 2019;

Bouvet et al., 2023). These observations are consistent with the peaks of sublimation/deposition obtained at the bottom/top boundaries when vapor is forced to no flux, as presented in Sect. 4.6.

These peaks of sublimation/deposition, and the related local increase of density gradients, are a trigger of a wave instability. This behaviour is carefully analysed in the study of Schürholt et al. (2022), who also discuss the implications on the numerical solution. The existence of such a wave instability was already proposed by Adams and Brown (1990) from theoretical argu-

ments. It is due to the dependence of the effective thermal conductivity and vapor diffusion coefficient to ice volume fraction. Physically, we understand the mechanism as follows: water vapor diffuses less readily across regions with higher ice volume fraction; because thermal conductivity of air is about 100 times less than that of ice, temperature gradients are stronger in regions with low ice volume fraction, and vapor fluxes are enhanced; these two phenomenon combine to induce local accumulation of water vapor in places where the latter encounter an abrupt increase in ice volume fraction in its flow towards

decreasing temperatures; this local accumulation leads to local over-saturation which is resorbed through deposition, increasing further the ice volume fraction at the transition. Similarly, when water vapor flows across an abrupt decrease of ice volume fraction, enhanced water fluxes in the downstream low-density region combined to the difficulty of the upstream high-density region to supply water vapor lead to local under-saturation, which causes sublimation. Our developments have highlighted two stabilising effects: one that is physical, i.e. the inclusion of settlement; the other that is numerical, i.e. the element-wise treat-

ment of $\Phi_i$. Given these two effects, and acknowledging the fact that the numerical set-up considered here is rather extreme (macroscopic temperature gradient of $1000$ $\mathrm{Km}^{-1}$, 1000 elements for a 2 $\mathrm{cm}$-thick snowpack), we think that this instability wave is unlikely to become a major modelling difficulty for many of the natural settings. This point is more hazardous for, e.g., seasonal-scale simulations of thin arctic snowpacks, and more simulations covering a large range of realistic configurations are needed.

As discussed by Simson et al. (2021), the surface ice mass balance related to new precipitation or to sublimation (for dry snow) has also to be addressed. We think that our choice regarding the element-wise treatment of conserved quantities will greatly facilitate this implementation. For simulations on a seasonal time scale, occasional re-meshing will be necessary to limit the computational cost. Again the element-wise nature of conserved quantities will enable unequivocal redistribution of the latter over merged or split elements. More generally, we draw the reader attention on the widespread confusion between

physical and numerical layers. Even when one claims adopting a continuous representation of the snowpack, the numerical solution of PDEs always require to discretize the domain, which amounts to the definition of numerical layers. As soon as some integration is at stake (e.g., assessment of snowpack deformation from stress and viscosity), some assumption must necessarily be done about the evolution of fields within the numerical layers. Therefore, it is obviously preferable to have finer mesh in areas where gradients of relevant physical quantities are strong, and a coarser mesh elsewhere. This is equivalent to say that





they must be some kind of correspondence between numerical and physical layers, if we define the latter as sections of the domain characterised by low gradients of physical properties.

The present work is a further step towards the development of the next generation of detailed snowpack models, associating sound and universal physics to a robust numerical treatment. So far, the implemented processes are limited to the coupled heat conduction, water vapor diffusion and settlement in dry snow, but the numerical subtleties that have been highlighted
are relevant for many other processes, in snow modelling and beyond. Our model is now ready to reproduce well-controlled laboratory experiments provided that BCs are well constrained. In contrast, modelling natural settings will require more work, in particular regarding the energy, water vapor mass, and ice mass balances at boundaries. For longer-term developments, it will be necessary to implement other processes, notably those related to the liquid phase of water (melting/refreezing, percolation), as well as those regarding snow metamorphism.

*Code availability.* The source files of the code are provided at https://doi.org/10.5281/zenodo.7941767 to guarantee the permanent reproducibility of results. However, we recommend that potential future users and developers access the code from its Git repository (https://github.com/jbrondex/ivori_model_homemadefem, last access: 16 May 2023) to benefit from the last versions of the code. The version used in this work is tagged as v0.1.0. For setting up the environment and running the simulations, please follow the instructions described in the README file present in the GitHub repository.

## Appendix A: Parameterizations used in our model

### A1 Effective vapor diffusion coefficient

All simulations presented in the paper are run using the parameterization of the effective vapor diffusion coefficient proposed by Simson et al. (2021). It is based on the parameterization derived by Calonne et al. (2014), but it uses the Heaviside function $\Theta$ to hinder vapor diffusion for ice volumes above two-thirds. It writes:

$$D^{\text{eff}} = D_0 \left(1 - \frac{3}{2}\Phi_{\text{i}}\right) \Theta\left(\frac{2}{3} - \Phi_{\text{i}}\right), \tag{A1}$$

with $D_0$ given in Table 1. Additional simulations mentioned in the paper were run using the parameterization of Hansen and Foslien (2015), which writes:

$$D^{\text{eff}} = \Phi_{\text{i}}(1 - \Phi_{\text{i}})D_0 + (1 - \Phi_{\text{i}}) \left( \frac{k_{\text{i}} D_0}{\Phi_{\text{i}}\left(k_{\text{a}} + L_{\text{m}} D_0 \frac{d\rho_{v,sat}}{dT} + (1 - \Phi_{\text{i}})k_{\text{i}}\right)} \right), \tag{A2}$$

with all constant values listed in Table 1.





## A2   Effective thermal conductivity

All simulations presented in the paper are run using the parameterization of Calonne et al. (2011) for the effective thermal conductivity, which writes:

$$k^{\text{eff}} = k_0 + k_1 \rho_i \Phi_i + k_2 (\rho_i \Phi_i)^2, \tag{A3}$$

with $k_0 = 0.024$ W m$^{-1}$ K$^{-1}$, $k_1 = -1.23 \times 10^{-4}$ W m$^2$ K$^{-1}$ kg$^{-1}$, and $k_2 = 2.5 \times 10^6$ W m$^5$ K$^{-1}$ kg$^{-2}$. Additional simulations mentioned in the paper were run using the parameterization of Hansen and Foslien (2015). The latter writes:

$$k^{\text{eff}} = \Phi_i \left( (1 - \Phi_i) k_a + \Phi_i k_i \right) + (1 - \Phi_i) \left( \frac{k_i k_a}{\Phi_i \left( k_a + L_m D_0 \frac{d\rho_{v,sat}}{dT} + (1 - \Phi_i) k_i \right)} \right), \tag{A4}$$

with all constant values listed in Table 1.

## A3   Effective heat capacity

Contrary to Simson et al. (2021) and Schürholt et al. (2022), our parameterization of the effective heat capacity neglects the heat carried by dry air, which then writes:

$$(\rho C_p)^{\text{eff}} = \rho_i C_i \Phi_i, \tag{A5}$$

with all constant values given in Table 1. This assumption is justified by the small volumetric heat capacity of dry air compared to that of ice. This choice enables to close the energy budget without having to track the dry air leaving the snowpack as it settles.

## A4   Saturation water vapor density

All simulations presented in this paper are run using the parameterization of Libbrecht (1999) for the saturation water vapor density:

$$\rho_v^{\text{eq}} = \frac{\exp(-T_r/T)}{fT} \left( a_0 + a_1 (T - T_m) + a_2 (T - T_m)^2 \right), \tag{A6}$$

with $T_r = 6150$ K, $f = 461.31$ J K$^{-1}$ kg$^{-1}$, $a_0 = 3.6636 \times 10^{12}$ Pa, $a_1 = -1.3086 \times 10^8$ Pa K$^{-1}$, $a_2 = -3.3793 \times 10^6$ Pa K$^{-2}$, and $T_m = 273$ K.

## A5   Effective viscosity

All simulations presented in the paper are run using the viscosity parameterization of Vionnet et al. (2012):

$$\eta(\Phi_i, T) = f \eta_0 \frac{\rho_i \Phi_i}{c_\eta} \exp \left( a_\eta (T_0 - T) + b_\eta \rho_i \Phi_i \right), \tag{A7}$$

with the fusion temperature $T_0 = 273$ K; the constant parameters $\eta_0 = 7.62237 \times 10^6$ kg s$^{-1}$, $a_\eta = 0.1$ K$^{-1}$, $b_\eta = 0.0.023$ m$^3$ kg$^{-1}$, $c_\eta = 250$ kg m$^{-2}$; and an additional parameter $f$ which normally accounts for snow microstructure properties but which is set to $f = 1$ in the present study.





## Appendix B: Discrete forms of the systems of equations

This appendix presents the different FEM discretizations of the Calonne and Hansen systems for all considered strategies that are summarized in Fig. 1. In what follows, the functions $\varphi_i$ and $\varphi_j$ correspond to, respectively, the test and shape functions.

### B1  CC_3DOF

The Calonne system is solved for the solution vector $\mathbf{u} = (T, \rho_{\mathrm{v}}, c)$. The discrete system evaluated at non-linear iteration $k+1$ can be expressed in matrix form as:

$$
\begin{cases}
M_{\mathrm{T}}\dot{T} + K_{\mathrm{T,T}}T + K_{\mathrm{T,C}}C & = F_{\mathrm{T}}^{\partial\Gamma} \\
M_{\mathrm{P}}\dot{P} + K_{\mathrm{P,P}}P + K_{\mathrm{P,C}}C & = F_{\mathrm{P}}^{\partial\Gamma} \\
\widetilde{M}_{\mathrm{C}}C + K_{\mathrm{C,T}}T + K_{\mathrm{C,P}}P & = F_{\mathrm{C}}
\end{cases}
\tag{B1}
$$

where:

$$
M_{\mathrm{T}}^{\mathrm{i,j}} = \int_{\Gamma} (\rho C_{\mathrm{p}})^{\mathrm{eff}}\,\varphi_i\varphi_j\mathrm{dV},
\tag{B2}
$$

$$
M_{\mathrm{P}}^{\mathrm{i,j}} = \int_{\Gamma} (1 - \Phi_{\mathrm{i}})\,\varphi_i\varphi_j\mathrm{dV},
\tag{B3}
$$

$$
\widetilde{M}_{\mathrm{C}}^{\mathrm{i,j}} = \int_{\Gamma} \varphi_i\varphi_j\mathrm{dV},
\tag{B4}
$$

$$
K_{\mathrm{T,T}}^{\mathrm{i,j}} = \int_{\Gamma} k^{\mathrm{eff}}\nabla\varphi_i \cdot \nabla\varphi_j\mathrm{dV},
\tag{B5}
$$

$$
K_{\mathrm{P,P}}^{\mathrm{i,j}} = \int_{\Gamma} D^{\mathrm{eff}}\nabla\varphi_i \cdot \nabla\varphi_j\mathrm{dV},
\tag{B6}
$$

$$
K_{\mathrm{T,C}}^{\mathrm{i,j}} = \int_{\Gamma} -L_{\mathrm{m}}\varphi_i\varphi_j\mathrm{dV},
\tag{B7}
$$

$$
K_{\mathrm{P,C}}^{\mathrm{i,j}} = \int_{\Gamma} \varphi_i\varphi_j\mathrm{dV},
\tag{B8}
$$





$$K_{C,T}^{i,j} = \int_\Gamma s\alpha v_{kin}(T^k) \left.\frac{d\rho_v^{eq}}{dT}\right|_{T=T^k} \varphi_i \varphi_j dV, \tag{B9}$$

$$K_{C,P}^{i,j} = \int_\Gamma -s\alpha v_{kin}(T^k)\varphi_i \varphi_j dV, \tag{B10}$$

$$F_C^i = \int_\Gamma s\alpha v_{kin}(T^k) \left( \left.\frac{d\rho_v^{eq}}{dT}\right|_{T=T^k} T^k - \rho_v^{eq}(T^k) \right) \varphi_i dV, \tag{B11}$$

where the saturation vapor density $\rho_v^{eq}$ has been linearized through Eq. (10), while $v_{kin}$ is fixed from the temperature field $T^k$ obtained at the previous non-linear iteration. Vectors $F_T^{\partial\Gamma}$ and $F_P^{\partial\Gamma}$ correspond to normal boundary fluxes of sensible heat and vapor respectively. Additional internal sources of energy beside phase change-related latent heat effect could easily be added as a force vector in the energy budget equation. Finally, note that the matrix $\widetilde{M}_C$ is not a true mass matrix in the sense that it does not apply to a time-derivative, and must therefore be treated carefully during the time-stepping assembly (9).

**B2   CC_2DOF**

The Calonne system is solved for the solution vector $\mathbf{u} = (T, \rho_v)$. The discrete system evaluated at non-linear iteration $k+1$ can be expressed in matrix form as:

$$\begin{cases} M_T \dot{T} + K_{T,T}T + K_{T,P}P & = F_T + F_T^{\partial\Gamma} \\ M_P \dot{P} + K_{P,P}P + K_{P,T}T & = F_P + F_P^{\partial\Gamma} \end{cases} \tag{B12}$$

where:

$$K_{T,T}^{i,j} = \int_\Gamma \left[ k^{eff}\nabla\varphi_i \cdot \nabla\varphi_j + L_m s\alpha v_{kin}(T^k) \left.\frac{d\rho_v^{eq}}{dT}\right|_{T=T^k} \varphi_i \varphi_j \right] dV, \tag{B13}$$

$$K_{P,P}^{i,j} = \int_\Gamma \left[ D^{eff}\nabla\varphi_i \cdot \nabla\varphi_j + s\alpha v_{kin}(T^k)\varphi_i \varphi_j \right] dV, \tag{B14}$$

$$K_{T,P}^{i,j} = \int_\Gamma -L_m s\alpha v_{kin}(T^k)\varphi_i \varphi_j dV, \tag{B15}$$

$$K_{P,T}^{i,j} = \int_\Gamma -s\alpha v_{kin}(T^k) \left.\frac{d\rho_v^{eq}}{dT}\right|_{T=T^k} \varphi_i \varphi_j dV, \tag{B16}$$





$$F_{\mathrm{T}}^{\mathrm{i}} = \int_{\Gamma} L_{\mathrm{m}} s\alpha v_{\mathrm{kin}}(T^k)\left(\left.\frac{d\rho_{\mathrm{v}}^{\mathrm{eq}}}{dT}\right|_{T=T^k} T^k - \rho_{\mathrm{v}}^{\mathrm{eq}}(T^k)\right)\varphi_i \mathrm{dV}, \tag{B17}$$

$$F_{\mathrm{P}}^{\mathrm{i}} = \int_{\Gamma} -s\alpha v_{\mathrm{kin}}(T^k)\left(\left.\frac{d\rho_{\mathrm{v}}^{\mathrm{eq}}}{dT}\right|_{T=T^k} T^k - \rho_{\mathrm{v}}^{\mathrm{eq}}(T^k)\right)\varphi_i \mathrm{dV}, \tag{B18}$$

while $M_{\mathrm{T}}$, and $M_{\mathrm{P}}$ are expressed as in the CC_3DOF case. Again, additional internal sources of energy could easily be added in the force vector $F_{\mathrm{T}}$ if needed, and the non-linearity due to $\rho_{\mathrm{v}}^{\mathrm{eq}}$ and $v_{\mathrm{kin}}$ is treated as for the CC_3DOF case. The field of $c$ is then diagnosed in a next step as the closure of the water vapor mass balance equation where $\rho_{\mathrm{v}}$ is set to the obtained solution field.

**B3  CD_PC**

The equations of the Calonne system are solved sequentially. In a first step, the heat equation is solved for $T$ with a source term that is fixed from the $c$-field computed at the previous time step. The discrete counterpart of this equation can be expressed in matrix form as:

$$M_{\mathrm{T}}\dot{T} + K_{\mathrm{T,T}}T = F_{\mathrm{T}} + F_{\mathrm{T}}^{\partial\Gamma}, \tag{B19}$$

where:

$$F_{\mathrm{T}}^{\mathrm{i}} = \int_{\Gamma} L_{\mathrm{m}} c\varphi_i \mathrm{dV}, \tag{B20}$$

and $M_{\mathrm{T}}$, $K_{\mathrm{T,T}}$ as expressed in the CC_3DOF case, and with the possibility to add additional internal sources of energy in $F_{\mathrm{T}}$. The obtained solution field is used to fix the distributions of $v_{\mathrm{kin}}$ and $\rho_v^{eq}$. The water vapor mass balance equation can then be solved for $\rho_{\mathrm{v}}$ under its diffusion-reaction form in a second step. The discrete counterpart of this equation can be expressed in 930 matrix form as:

$$M_{\mathrm{P}}\dot{P} + K_{\mathrm{P,P}}P = F_{\mathrm{P}} + F_{\mathrm{P}}^{\partial\Gamma} \tag{B21}$$

where:

$$F_{\mathrm{P}}^{\mathrm{i}} = \int_{\Gamma} s\alpha v_{\mathrm{kin}}(T^{k+1})\rho_{\mathrm{v}}^{\mathrm{eq}}(T^{k+1})\varphi_i \mathrm{dV}, \tag{B22}$$

and $M_{\mathrm{P}}$, $K_{\mathrm{P,P}}$ as expressed in the CC_2DOF case. We stress that, because $v_{\mathrm{kin}}$ and $\rho_v^{eq}$ are fixed from the field $T^{k+1}$ obtained 935 in a previous (separated) step, no linearization is required in this case. The field of $c$ is computed in a third step as the closure of the water vapor mass balance equation where $\rho_{\mathrm{v}}$ is set to the obtained solution field.





## B4 CD_FD

The CD_FD strategy is similar to the CD_PC strategy except that the $c$-field computed at the previous time step is used to fix the source terms of both the heat and water vapor mass balance equations. It follows that the discrete counterparts of both

equations write exactly the same as for the CD_PC case, except for $K_{\mathrm{P,P}}$ and $F_{\mathrm{P}}$ which write:

$$K_{\mathrm{P,P}}^{\mathrm{i,j}} = \int_{\Gamma} D^{\mathrm{eff}} \nabla \varphi_i \cdot \nabla \varphi_j \mathrm{dV}, \tag{B23}$$

and

$$F_{\mathrm{P}}^{\mathrm{i}} = \int_{\Gamma} -c \varphi_i \mathrm{dV}. \tag{B24}$$

As for the CD_PC case, no linearization is required in this approach. The field of $c$ is computed in a third step by solving

Eq. (2) at all nodes, where $\rho_{\mathrm{v}}$ and $\rho_{\mathrm{v}}^{\mathrm{eq}}(T)$ are set to the obtained solution fields.

## B5 H_MF

In the Hansen system, noting $H$ the vector of enthalpy content, the total (i.e. sensible plus latent) energy budget writes in matrix form:

$$\begin{cases} M_{\mathrm{H}} \dot{H} + K_{\mathrm{T,T}} T &= F_{\mathrm{T}}^{\partial \Gamma} \\ \widetilde{M}_{\mathrm{H}} H + K_{\mathrm{H,T}} T &= F_{\mathrm{H}} \end{cases} \tag{B25}$$

where:

$$M_{\mathrm{H}}^{\mathrm{i,j}} = \int_{\Gamma} \varphi_i \varphi_j \mathrm{dV}, \tag{B26}$$

$$\widetilde{M}_{\mathrm{H}}^{\mathrm{i,j}} = \int_{\Gamma} \varphi_i \varphi_j \mathrm{dV}, \tag{B27}$$

$$K_{\mathrm{T,T}}^{\mathrm{i,j}} = \int_{\Gamma} \left( k^{\mathrm{eff}} + D^{\mathrm{eff}} L_{\mathrm{m}} \left. \frac{d\rho_{\mathrm{v}}^{\mathrm{eq}}}{dT} \right|_{T=T^k} \right) \nabla \varphi_i \cdot \nabla \varphi_j \mathrm{dV}, \tag{B28}$$

$$K_{\mathrm{H,T}}^{\mathrm{i,j}} = \int_{\Gamma} -\left( L_{\mathrm{m}}(1-\Phi_{\mathrm{i}}) \left. \frac{d\rho_{\mathrm{v}}^{\mathrm{eq}}}{dT} \right|_{T=T^k} + (\rho C_{\mathrm{p}})^{\mathrm{eff}} \right) \varphi_i \varphi_j \mathrm{dV}, \tag{B29}$$

$$F_{\mathrm{H}}^{\mathrm{i}} = \int_{\Gamma} \left( L_{\mathrm{m}}(1-\Phi_{\mathrm{i}})(\rho_{\mathrm{v}}^{\mathrm{eq}}(T^k) - \left. \frac{d\rho_{\mathrm{v}}^{\mathrm{eq}}}{dT} \right|_{T=T^k} T^k) - (\rho C_{\mathrm{p}})^{\mathrm{eff}} T_0 \right) \varphi_i \mathrm{dV}, \tag{B30}$$





where the saturation vapor density $\rho_{\mathrm{v}}^{\mathrm{eq}}$ has been linearized through Eq. (10). Vector $F_{\mathrm{T}}^{\partial\Gamma}$ corresponds to the normal heat flux (sensible and latent) at the boundary. This matrix representation is referred to as "mixed-form" as the heat budget equation includes two unknowns, the total energy $H$ and the temperature $T$, which are related through a non-linear constitutive equation. As for the CC_3DOF case, note that the matrix $\widetilde{M}_{\mathrm{H}}$ is not a true mass matrix and must be treated carefully during the time-
stepping assembly (9). The field of $c$ is then diagnosed in a next step as the closure of the water vapor mass balance equation with $\rho_{\mathrm{v}} = \rho_v^{eq}$, where $\rho_v^{eq}$ is computed from the obtained field of $T$.

**B6   H_TF**

Applying the chain rule in the Hansen system of equations, one can eliminate the total energy $H$ to express the system in terms of $T$ only. The equivalent matrix form is given by:

$$M_{\mathrm{T}}\dot{T} + K_{\mathrm{T,T}}T = F_{\mathrm{H}}^{\partial\Gamma} \tag{B31}$$

where

$$M_{\mathrm{T}}^{\mathrm{i,j}} = \int\limits_{\Gamma}\left((\rho C_{\mathrm{p}})^{\mathrm{eff}} + L_{\mathrm{m}}(1-\Phi_{\mathrm{i}})\left.\frac{d\rho_{\mathrm{v}}^{\mathrm{eq}}}{dT}\right|_{T=T^k}\right)\varphi_i\varphi_j\mathrm{dV}, \tag{B32}$$

and the expression of $K_{\mathrm{T,T}}$ is the same as for the H_MF case. Again, the field of $c$ is then diagnosed in a next step as the closure of the water vapor mass balance equation with $\rho_{\mathrm{v}} = \rho_v^{eq}$, where $\rho_v^{eq}$ is computed from the obtained field of $T$.

**Appendix C:  Modifications made to the published code of Simson et al. (2021)**

After running the code published by Simson et al. (2021) (last access: 2023-05-04), three bugs (i.e., inconsistencies in the code that hamper mass conservation) have been identified. Here, we detail these three bugs and explain the changes that were made to the lines of code that caused the problems to fix these bugs. In addition to these bugs, we recall that the explicit temporal discretization of the continuity equation performed by Simson et al. (2021) also causes a violation of mass conservation (Sect.
3.3 and Appendix D). Although we consider this last point to be of a conceptual order rather than a bug (the problem appears both in the paper and in the code), we also present the modifications that are needed in the published code to switch from the original explicit time discretization to the implicit one.

**C1   Spatial inconsistency in the strain rate computation**

We recall that the settlement scheme consists in two operations that must be done consistently to guarantee mass conservation
(Sect. 3.3): (i) the update of the ice volume fraction field, and (ii) the update of the mesh. The strain rate is required in both operations. In the approach of Simson et al. (2021), the strain rate is evaluated at nodes: the strain rate at node $k$ is calculated from the viscosity and stress at node $k$. As explained in Sect. 3.3, the stress at node $k$ is calculated from the overburden




snow mass with the implicit assumption that the ice volume fraction $\Phi_{i,k}$ stored at node $k$ actually applies to the whole space interval located above, i.e. between nodes $k$ and $k+1$. On the other hand, in the published version of the code, the authors

were updating the mesh using the strain rate evaluated at node $k$ to compute the deformation of the space interval below, i.e. between the nodes $k$ and $k-1$. This operation was done in the file `Model/velocity.py` through the following code lines (l.214-220):

```
D_rate[
    0
] = 0 # strain rate at lowest node = 0, intersection with the ground no strain
v[0] = D_rate[0] * dz[0] # local velocity at the lowest node v=0 [ms-1]
v[1:] = np.cumsum(
    D_rate[1:] * dz[:]
) # Integrate deformation rates [s-1] in space to derive velocity [ms-1]
```

The fact that the ice volume fraction stored at node $k$ relates to the snow mass contained in the element above, whereas the strain rate calculated at node $k$ is used for the deformation of the element below, leads to inconsistencies when ice volume fraction is updated. Therefore, we modified the code so that the strain rate used to deform the space interval between nodes $k$ and $k-1$ during the mesh update step is the one evaluated at node $k-1$ instead of the one evaluated at node $k$. Concretely, the code lines reported above are replaced by:

```
v[0] = 0 # local velocity at the lowest node v=0 [ms-1]
        v[1:] = np.cumsum(
            D_rate[:-1] * dz[:]
        ) # Integrate deformation rates [s-1] in space to derive velocity [ms-1]
```

**C2  Inconsistency in the sequence of tasks**

In the published version of the code, the strain rate was updated in between the ice volume fraction update and mesh update steps, leading to inconsistency between the two operations. This inconsistency was corrected by replacing the following code lines of the file `Model/coupled_update_phi_coord.py` (l.13-16):

```
phi_new = update_phi(c, dt, nz, phi, v_dz)
(coord_new, dz, v_dz_new, v_new, sigma) = update_coord(
    T, c, dt, nz, phi, coord, SetVel, v_opt, viscosity
)
```

by:

```
(coord_new, dz, v_dz_new, v_new, sigma) = update_coord(
    T, c, dt, nz, phi, coord, SetVel, v_opt, viscosity
)
```





```
phi_new = update_phi(c, dt, nz, phi, v_dz_new)
```

### C3   Inconsistency in the relationship between $\Phi_i$ and the effective snow density

In the code published by Simson et al. (2021), which is consistent with Eq. (18) of their article, the stress is calculated from the overburden snow mass neglecting the contribution of the air mass, i.e. assuming the effective snow density is $\rho_{snow} = \rho_i \Phi_i$.

However, in the code, the initial field of ice volume fraction $\Phi_i^0$ is retrieved from the user-prescribed initial field of snow density $\rho_{snow}^0$ as:

$$\Phi_i^0 = \frac{\rho_{snow}^0 - \rho_a}{\rho_i - \rho_a},\tag{C1}$$

where $\rho_a = 1.335\,\mathrm{kg\,m^{-3}}$ is the air density. To correct this inconsistency, line 19 of the file `Model/phi_from_rho_eff.py`, which was:

```
phi = np.true_divide((rho_eff - rho_a), (rho_i - rho_a))
```

is replaced by:

```
phi = np.true_divide(rho_eff, rho_i)
```

### C4   Explicit versus implicit time integration of the continuity equation

In the published version of the code, the continuity equation is discretized using an explicit time integration scheme. This is

done in the file `Model/coupled_update_phi_coord.py`, through the following code line :

```
phi_new = phi + dt * (c / rho_i - v_dz * phi)
```

Switching to the mass-conservative implicit time integration scheme implies to replace this code line by the following:

```
phi_new = (phi + dt * (c / rho_i))/(1 + dt * v_dz)
```

### Appendix D:  Mass-conservative temporal discretization of the continuity equation

Let consider the numerical layer $k+1/2$ undergoing settlement without any phase change between $t$ and $t+\Delta t$. If we denote $\Phi_{i,k+1/2}^n$ and $L_{k+1/2}^n$ (resp. $\Phi_{i,k+1/2}^{n+1}$ and $L_{k+1/2}^{n+1}$) the ice volume fraction and length of the numerical layer before (resp. after) settlement, and if we neglect the contribution of air in the snow mass, then the mass conservation simply writes:

$$\Phi_{i,k+1/2}^{n+1} L_{k+1/2}^{n+1} = \Phi_{i,k+1/2}^n L_{k+1/2}^n,\tag{D1}$$

which can be rewritten:

$$\Phi_{i,k+1/2}^{n+1} = \Phi_{i,k+1/2}^n \frac{L_{k+1/2}^n}{L_{k+1/2}^{n+1}}.\tag{D2}$$



The total deformation of the layer $k+1/2$ over $\Delta t$ is:

$$L_{k+1/2}^{n+1} - L_{k+1/2}^n = \int\limits_{z=z_k^n}^{z_{k+1}^n} \dot{\epsilon}_{zz}^{n+1}(z)dz\Delta t, \tag{D3}$$

and thus the spatially averaged strain writes:

$$\frac{L_{k+1/2}^{n+1} - L_{k+1/2}^n}{L_{k+1/2}^n} = \frac{\int_{z=z_k^n}^{z_{k+1}^n} \dot{\epsilon}_{zz}^n(z)dz}{L_{k+1/2}^n}\Delta t = \dot{\epsilon}_{zz,k+1/2}^{n+1}\Delta t. \tag{D4}$$

It follows that:

$$\frac{L_{k+1/2}^{n+1}}{L_{k+1/2}^n} = 1 + \dot{\epsilon}_{zz,k+1/2}^{n+1}\Delta t. \tag{D5}$$

Combining Eqs (D2) and (D5) gives:

$$\Phi_{i,k+1/2}^{n+1} = \frac{\Phi_{i,k+1/2}^n}{1 + \dot{\epsilon}_{zz,k+1/2}^{n+1}\Delta t}. \tag{D6}$$

Equation (D6) guarantees mass conservation by construction (provided that the air contribution in the snow mass is neglected).
Equation (21) is an extension of Eq. (D6) which also includes the contribution of the mean deposition within layer $k+1/2$.
When phase change is present, the deposition/sublimation effectively changes the ice volume fraction affected to the layer.

*Author contributions.* MD obtained funding and supervised the work. JB and FT made the state of the art. JB implemented the model with
constant support of KF, and insights from MD, NC, PH and HL. JB ran the numerical simulations. JB interpreted the results with help from
all co-authors. JB wrote the manuscript with contributions from all co-authors.

*Competing interests.* The authors declare that they have no conflict of interest.

*Acknowledgements.* We thank Clément Cancès for fruitful discussions on numerical methods. We thank Louis Le Toumelin for his help in
managing the git repository. CNRM/CEN is part of Labex OSUG (ANR10 LABX56). JB, KF, MD, and FT have received funding from the
European Research Council (ERC) under the European Union's Horizon 2020 research and innovation program (IVORI (grant no. 949516)).



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
