# Peer review of "A finite-element framework to explore the numerical solution of the coupled problem of heat conduction, water vapor diffusion and settlement in dry snow (IvoriFEM v0.1.0)"

_Geoscientific Model Development, 2023_

## Author Comment (AC1)

**Referee #1**

First, we would like to thank the anonymous referee #1 for her/his careful reading and her/his highly detailed specific comments on our paper.

**General comments**

This is a thorough investigation of numerics for simulation of coupled processes in snow. As such, it is not an easy read and is very long. I wonder if it might be shortened beneficially. For example, the description of FEM will be unnecessary for those already familiar with it while not providing everything needed for understanding by those who are not. Otherwise, I have just reproduced editorial corrections that I noted in the manuscript; these will be of no interest for readers of the discussion, but should help the authors to improve the paper.

We agree that the first part of Section 3.1 (i.e., the part about the spatial and temporal discretization) is rather general, and we have therefore moved it to the Appendix. For what regards the remaining of the paper, although it is admittedly rather long and technical, our opinion is that none of the given explanations are superflous. First, we wanted to unequivocally describe the specifities of our implementation compared to the ones of Schürholt et al. (2022) and Simson et al. (2021). Second, our paper deals with issues that are of prime importance for proper modeling of snowpacks evolution. The soil/snow modeling community is therefore our targeted audience, although some of our results are relevant beyond this scope. Scientists belonging to this community (just like us) usually use numerical methods as a tool, and specific issues falling in the realm of applied mathematics, such as the ones we are investigating in this work, seldom get the careful attention they deserve. This can lead to misunderstandings or inadequate numerical practices that spread from publication to publication. For example, numerical modellers are not always aware that a Crank-Nicholson scheme can be affected by spurious (decaying) oscillations despite its unconditional L2 stability if the time step is too large, or that linearization of non-linear systems can put a constrain on the time step requirement even when using a fully-implicit scheme. Therefore, we wrote our paper with the constant concern of giving enough details so that all explanations are self-sufficient for a reader with a background in numerical modelling of earth system components.

**Specific comments**

Line 12: delete "to get" → Done.

Line 33: "echoes on" is a curious choice of words → It has been replaced by "feeds back on".

Line 51: "too large relative to" → Corrected.

Line 72: None of these references are to firn intercomparison projects. The statement is, I think, true for snow intercomparison projects. Mass and energy conservation was checked in earlier land surface model intercomparison projects (e.g. https://www.sciencedirect.com/science/article/pii/S0921818198000447); it was a surprise at the time that models did not all balance. Non-conservation and consequent drift is of more concern and more apparent in coupled climate models. → Thank you for these details. We removed "firn" from the text. Note that the question of non-conservation and related drift in coupled climate models is briefly discussed in Section 5 of our paper.

Line 90: "is a an opening" - delete "a"

Line 108: "the macroscopic system of equations is"

Line 123: "dependency of the effective parameters on T"

The three points listed above have been corrected.

Line 129: https://doi.org/10.1029/2010EO450004 → We have replaced parentheses by slash ponctuation marks.

Line 139: delete "which then writes"

Line 141: "consists of a single prognostic equation for T"

Line 158: "lies in the fact"

Line 207: "we code every step of the method internally"

Line 227: "that can be cast"

Line 237: "the matrix form of the resulting algebraic system is"

Line 250: "while every case ... is first-order in time"

Line 253: "combined with its stability"

Line 264: "it is possible to diagnose the fluxes"

Line 270: "One of the main goals"

Line 271: "Details of the various approaches ... to reach this conclusion"

Line 285: delete "what regards"

Line 334: "we refer to this as the proper form of the mass matrix"

Line 348: "this method enables removing spurious oscillations"

Line 383: delete "imperatively"

Line 398: "obtained at the previous time step, is equivalent to performing"

Line 399: "thus does not necessarily"

Line 404: "without any reference to how these fluxes"

Line 425: "this allows eliminating ... from the continuity equation, which is then"

Line 447: "the mass contained within each layer remains the same"

Line 452: "would enable switching"

Line 457: "consists of two numerical operations"

Line 471: "without any explicit reference to how these fields should vary between nodes"

Line 484: "an inconsistency that hampers mass conservation"

Line 487: delete "which writes"

Line 490: delete "comprised"

Line 515: no need to abbreviate "respectively"

All points listed above have been corrected following your recommendations.

Figure 3: If the dashed blue "No Lump.; Prop." line were plotted on top of the solid orange "Lump.; Prop.", then both could be made visible (like "FEniCS" and "No Lump.; Improp.").

We tried to modify Figure 3 according to your suggestion. The resulting figure is reported at the end of this document. It is true that this choice enables to spot both the cases "Lump.; Prop." and "No Lump.; Prop." on the top and middle panels. However, it makes the bottom panel more confused. The important result of this section is that lumping the mass matrix enables to remove spurious oscillations on the c-field. We think that this result is more obvious with the original version of the figure than with this one. Therefore, we have decided to keep the figure as it was.

Line 543: "treated in detail"

Line 556: "deviation in c"

Line 565: delete "of"

All points listed above have been corrected following your recommendations.

Line 613: delete "what regards" $\rightarrow$ We find the original formulation clearer.

Line 625: "associated with stronger deposition rates"

Line 629: "implies solving"

Line 674: "equivalent to imposing"

Line 685: "This oscillatory pattern propagates inwards over a number of nodes that grows in time."

Line 686: "but does not trigger oscillations"

Line 708: "into an ice"

Line 733: "associated with continuous sublimation"

Line 737: "the dependence of $k_{\text{eff}}$ and $D_{\text{eff}}$ on ice volume fraction"

Line 744: delete "of"

Line 753: delete "of", twice

Line 769: "This is a generic result"

Line 772: "might require considerably decreasing the time step"

Line 778: "coming up with"

Line 780: "This required knowing"

Line 783: "associated with"

Line 824: "the reader's attention to"

Line 825: "Even when one claims to be adopting"

Line 826: "always requires discretization of the domain"

Line 828: "must necessarily be made"

Line 829: "This is equivalent to saying that there must"

Line 833: "with a robust numerical treatment"

Line 868: "enables closing"

All points listed above have been corrected following your recommendations.

Line 971: A link to https://zenodo.org/record/5588308 would be useful here in addition to reference to Simson et al. (2021). → Good point. The link has been added.

Line 979: "consists of" → Corrected.

**Referee #2**

First, we would like to thank the anonymous referee #2 for her/his insightful comments on our paper.

**General comments**

This paper presents a valuable scientific contribution that advances the field by integrating water vapor transport with heat conduction and settling in dry firn models. The approach is particularly good for its modular program design, enabling potential extensions in the future. The utilization of mass conservative mixed forms of partial differential equations (PDEs) adds to the paper's robustness. The codes are well commented and the readme to the GitHub repository is clear to understand.

Addressing the below-mentioned points may refine the presentation and contribute to the overall impact of this work.

Major points:

**1. Streamlining Content:** The paper contains sections that come across as repetitive and overly verbose, potentially obscuring the core scientific contributions. It might be beneficial to consider either omitting or relocating such sections to the Appendix. For instance, the detailed description of finite element methods could be moved to appendix.

This point has also been raised by Referee #1. Please refer to the answer that has been done to her/his comment above.

**2. Enhancing Clarity:** The description of the implementation of boundary conditions, such as the Robyn type, could be further elaborated upon to ensure greater clarity and ease of repeatability. A more detailed explanation would contribute to the paper's overall accessibility.

A few sentences have been added to the "Spatial and temporal discretization" section (which is now Appendix B) to clarify this point. Robin boundary conditions can be seen as a weighted combination of Dirichlet and Neumann BCs. Very generally, if we denote $u$ the solution field of a PDE on a domain $\Omega$, then a Robin BC on the domain boundary $\partial\Omega$ has the form:

$$au + b\frac{\partial u}{\partial n} = f \ \text{ on } \partial\Omega. \tag{1}$$

In the numerical discretization step, the flux term $b\partial_n u$ naturally goes into the boundary flux vector $\mathbf{F}^{\partial\Gamma}$ as for a Neumann BC. Then, the line corresponding to the considered boundary of the l.h.s. matrix $\mathbf{A}$ must be replaced by a line of zeros, except for the diagonal term which must be replaced by the multiplying factor $a$ (similarly to what would be done for a Dirichlet BC, but with $a$ instead of 1 on the diagonal). The r.h.s. vector $\mathbf{B}$ is left unchanged. Note that in the case where $a$ depends on the solution $u$, the BC introduces a non-lineary in the system. This non-linearity is treated in the linearization loop through a Newton or a Picard method as for any other non-linearity.

**3. Tolerance Criterion Discussion:** The choice of the tolerance criterion (as shown in Equation 11) plays a pivotal role in achieving accuracy. It would be valuable to discuss the criticality of this choice in relation to Equation 361, where readers could better grasp its significance.

We have added a few lines to the manuscript to discuss this point. The convergence criterion that we have implemented corresponds to the one that is used by default in the finite element suite Elmer. Elmer is an open-source multi-physics code that is very popular among glaciologists (Gagliardini et al., 2013), and with which several authors of the present study are familiar. A number of other measures of the change of the solution between two consecutive iterations could be used as the convergence criterion. One of the most commonly adopted is the following:

$$2\frac{\left|\mathbf{u}^{k+1} - \mathbf{u}^k\right|}{|\mathbf{u}^{k+1}| + |\mathbf{u}^k|} < \varepsilon. \tag{2}$$

The convergence criterion is used only to stop the non-linear iterations, and it has no impact on the intermediate iterates. In other word, if the linearization algorithm converges, then it converges towards the same solution independently of the chosen criterion. Nevertheless, it is possible to think about situations in which, at a given iteration, certain convergence criteria would be met (and the linearization algorithm would thus stop), while others would not (and the linearization algorithm would go to a next iteration). For example, if two consecutive iterates have the exact same norm but are not the same, the convergence criterion used in our paper would be satisfied while the criterion (2) might not. In all simulations that are presented in the paper, the systems at stake are not strongly non-linear and only 1 to 3 non-linear iterations are needed to satisfy the implemented convergence criterion for $\epsilon = 10^{-5}$. Most importantly, the convergence is smooth, and the solution is already very close to its final (converged) value after the first non-linear iteration (i.e., the change of the solution between, e.g., iteration 1 and 2 is very small compared to the change between first guess and iteration 1). Therefore, we think that changing the implemented convergence criterion for, e.g., the criterion (2) would only marginally affect the obtained solution fields.

To gain confidence on this point, we decided to re-run all simulations presented in Section 4.2 (which gathers the cases CC_3DOF, CC_2DOF, CD_PC, H_MF and H_TF) changing the convergence criterion for the criterion (2). Calculating RMSDs, we compare the obtained $T$, $\rho_v$, and $c$-fields to the ones that were obtained with the original convergence criterion. After 2 h of simulation, all solution fields obtained with the new criterion are strictly the same as what they were with the original criterion, independently of the considered time step. After 24 h of simulation, all solution fields obtained with the new criterion are strictly the same as what they were with the original criterion when the time step is $\Delta t = 15$ min. This continues to be the case when the time step is decreased to $\Delta t = 5$ min for the approaches H_MF and CD_PC. Note that this was expected for the CD_PC approach as the non-linearity vanishes in this case (see manuscript). In contrast, tiny differences are observed on the solution fields produced with the CC_3DOF, CC_2DOF, and H_TF approaches after 24 h of simulation when $\Delta t = 5$ min. The corresponding RMSDs are reported in Table 1. These RMSDs are, in the worst cases, 3 to 4 orders of magnitude lower than the ones reported in the manuscript when comparing the different numerical approaches with each other. In other words, our results show very little sensitivity to the formulation of the convergence criterion (at least for the two criteria tested), and our conclusions remain valid even when adopting the convergence criterion (2) instead of the one we have adopted. Having said that, the point you are raising is a very relevant one, and future model developments as part of the ERC IVORI will surely benefit from a deeper reflection on the most suited convergence criterion to adopt.

Table 1: RMSDs between solutions obtained with the two tested convergence criteria after 24 h of simulation with $\Delta t = 5$ min.

| RMSD on: | $T$ (K) | $\rho_\mathrm{v}$ (kg m$^{-3}$) | $c$ (kg m$^{-3}$ s$^{-1}$) |
|---|---|---|---|
| CC_3DOF | $4.6 \times 10^{-9}$ | $6.1 \times 10^{-13}$ | $9.8 \times 10^{-14}$ |
| CC_2DOF | $6.8 \times 10^{-7}$ | $6.3 \times 10^{-11}$ | $1.0 \times 10^{-10}$ |
| CD_PC | $0$ | $0$ | $0$ |
| H_MF | $0$ | $0$ | $0$ |
| H_TF | $4.5 \times 10^{-6}$ | $9.0 \times 10^{-10}$ | $2.0 \times 10^{-12}$ |

**4. Addressing Validity of Linear Flow Law:** The paper could benefit from a concise discussion regarding the validity of the linear flow law for Newtonian fluids, particularly at short time scales (as mentioned in lines #430-438). It would be beneficial to cite relevant work, such as Simpson et al. 2021, to provide context and substantiate the argument.

Deriving a physically-based effective rheological law of snow that would describe the viscous deformation of a snowpack at the macro-scale from physical processes occurring at the micro-scale is still an unresolved problem of snow science (e.g., Fourteau et al., 2023). Thus, formulations of snow compaction in models are largely empirical. The two reference detailed snowpack models CROCUS and SNOWPACK make use of a linear flow law (Bartelt and Lehning, 2002; Vionnet et al., 2012), and we follow them. Discussing the validity of this law is out of the scope of this work. Indeed, as stated in lines 435-436 of the submitted manuscript: "the goal of this study is not an assessment of the model sensitivity to parameterization choices but to the details of the numerical treatment of the equations". In particular, considering a non-linear flow law would not change the conclusions of our paper regarding the proper numerical treatment to adopt for the settlement scheme in order to ensure mass conservation. The stabilising effect of snow compaction on the density wave instabilities that we have highlighted is also independent of the form of the flow law. In addition, we want to stress that due to the modularity of our code, the adopted linear flow law could be easily replaced by a non-linear one. For all these reasons, we have left the manuscript unchanged on this point.

**Specific comments**

Line 90: a an → an

Line 251, 253, 256: Cranck → Crank

Line 526: Full-implicit → fully implicit

Line 261: l.h.s → l.h.s.

Line 263: r.h.s → r.h.s.

All points listed above have been corrected.

**References**

Bartelt, P. and Lehning, M.: A physical SNOWPACK model for the Swiss avalanche warning: Part I: numerical model, Cold Regions Science and Technology, 35, 123–145, https://doi.org/10.1016/S0165-232X(02)00074-5, 2002.

Fourteau, K., Freitag, J., Malinen, M., and Löwe, H.: Microstructure-based simulations of the viscous densification of snow and firn, EGUsphere, 2023, 1–26, https://doi.org/10.5194/egusphere-2023-1928, 2023.

Gagliardini, O., Zwinger, T., Gillet-Chaulet, F., Durand, G., Favier, L., de Fleurian, B., Greve, R., Malinen, M., Martín, C., Råback, P., Ruokolainen, J., Sacchettini, M., Schäfer, M., Seddik, H., and Thies, J.: Capabilities and performance of Elmer/Ice, a new-generation ice sheet model, Geoscientific Model Development, 6, 1299–1318, https://doi.org/10.5194/gmd-6-1299-2013, 2013.

Schürholt, K., Kowalski, J., and Löwe, H.: Elements of future snowpack modeling - Part 1: A physical instability arising from the nonlinear coupling of transport and phase changes, The Cryosphere, 16, 903–923, https://doi.org/10.5194/tc-16-903-2022, 2022.

Simson, A., Löwe, H., and Kowalski, J.: Elements of future snowpack modeling–Part 2: A modular and extendable Eulerian–Lagrangian numerical scheme for coupled transport, phase changes and settling processes, The Cryosphere, 15, 5423–5445, https://doi.org/https://doi.org/10.5194/tc-15-5423-2021, 2021.

Vionnet, V., Brun, E., Morin, S., Boone, A., Faroux, S., Le Moigne, P., Martin, E., and Willemet, J. M.: The detailed snowpack scheme Crocus and its implementation in SURFEX v7.2, Geoscientific Model Development, 5, 773–791, https://doi.org/10.5194/gmd-5-773-2012, 2012.

[Figure]

Figure 1: Modification of Figure 3 so that the dashed blue "No Lump.; Prop." lines are plotted on top of the solid orange "Lump.; Prop." lines.

---

## Author Response (AR2)

Dear Editor,

Thank you for your comments on our paper. Below you will find the answers to the minor points you raised.

- Section 3.1: after re-organising the text, it seems that PDE and FEM are no longer defined. Please also check if other issues are still present due to text removal.

The acronym PDE (partial differential equation) was already defined at its first occurrence (l. 172). The acronym FEM (finite-element method) was indeed not defined. It is now defined at its first occurrence (l. 198). We have checked that text removal did not introduce any other issues.

- Many sub- and superscripts in your equation variables are in italic, which should not be. Please keep as italic only variables, and set all other names to straight text (e.g. Eq. 7 - ice and further). Please verify and adapt all occurrences that need a fix.

Done.

- Please add a bibliography entry for the Zenodo link of your code, and refer to it in the manuscript (in the main text and/or in the code availability section)

Done.

Sincerely,

Julien Brondex for all the authors